# Light of Normals: Unified Feature Representation for Universal Photometric Stereo

**Houyuan Chen**[1*]   **Hong Li**[2,3*]   **Chongjie Ye**[3,4†]   **Zhaoxi Chen**[5]   **Bohan Li**[3]

Shaocong Xu[3]   Xianda Guo[6]   Xuhui Liu[2]   Yikai Wang[7]   Baochang Zhang[2‡]

Satoshi Ikehata[8]   Boxin Shi[9]   Anyi Rao[1]   Hao Zhao[3,10‡]

[1]HKUST   [2]BUAA   [3]BAAI   [4]FNii, CUHKSZ
[5]NTU   [6]WHU   [7]BNU   [8]NII   [9]PKU   [10]AIR, THU

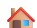 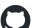 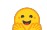

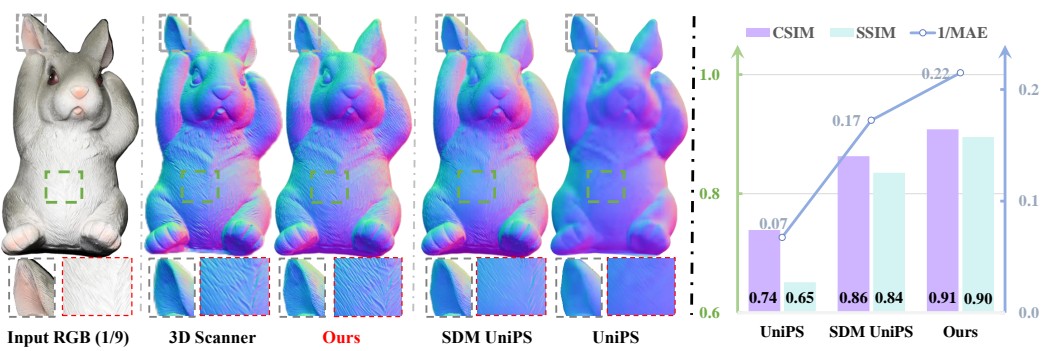

Figure 1: (left) Given multi-light images from a fixed viewpoint, **LINO UniPS** recovers sharper, more faithful normals than UniPS/SDM UniPS and visually rivals a 3D scanner. (right) On the DiLiGenT, a clear correlation exists between the consistency of encoder features (CSIM/SSIM) and the final reconstruction accuracy (1/MAE).

## Abstract

Universal photometric stereo (PS) is defined by two factors: it must (i) operate under arbitrary, unknown lighting conditions and (ii) avoid reliance on specific illumination models. Despite progress (e.g., SDM UniPS), two challenges remain. **First**, current encoders cannot guarantee that illumination and normal information are decoupled. To enforce decoupling, we introduce LINO UniPS with two key components: (i) Light Register Tokens with Light Alignment supervision to aggregate point, direction, and environment lights; (ii) Interleaved Attention Block featuring global cross-image attention that takes all lighting conditions together so the encoder can factor out lighting while retaining normal-related evidence. **Second**, high-frequency geometric details are easily lost. We address this with (i) a Wavelet-based Dual-branch Architecture and (ii) a Normal-gradient Perception Loss. These techniques yield a **unified** feature space in which lighting is explicitly represented by register tokens, while normal details are preserved via wavelet branch. We further introduce PS-Verse, a large-scale synthetic dataset graded by geometric complexity and lighting diversity, and adopt curriculum training from simple to complex scenes. Extensive experiments show new state-of-the-art results on public benchmarks (e.g., DiLiGenT, Luces), stronger generalization to real materials, and improved efficiency; ablations confirm that Light Register Tokens + Interleaved Attention Block drive better feature decoupling, while Wavelet-based Dual-branch Architecture + Normal-gradient Perception Loss recover finer details.

---

*Equal contribution, order interchangeable.

†Part of project lead.

‡Corresponding author.

# 1 INTRODUCTION

Photometric stereo (PS) Woodham (1980) aims to recover surface normals from multiple images under varying lighting conditions. Traditional methods Ikehata et al. (2012; 2014); Ikehata (2018); Mo et al. (2018); Haefner et al. (2019) rely on assumptions of specific light sources or physical models, making them difficult to adapt to complex natural scenes.

Recent Universal PS methods Ikehata (2022; 2023); Hardy et al. (2024) have alleviated this reliance by replacing explicit physical modeling with a learnable encoder-decoder architecture. The encoder extracts representations of pixel-wise, spatially varying illumination from multiple input images and produces a global lighting context, while the decoder, acting as a calibration network Ikehata (2021), transforms this context into normal predictions. This paradigm achieves robust normal estimation in diverse and spatially non-uniform lighting environments, substantially advancing the field. Yet, achieving reliable performance in this setting is more challenging than it seems.

The first challenge lies in the **ineffective decoupling of illumination and normal cues**. Existing encoders jointly process lighting and normal, but without explicit illumination representation the decoder inherits unstable features, producing inconsistent normal predictions. Interestingly, we observe that when encoders produce more *similar* normal features across different inputs, the accuracy of normal estimation improves (Fig. 1 right), suggesting that explicitly factoring out illumination is crucial for normal recovery. The second challenge is the **loss of geometry details**. Conventional up/downsampling operations (e.g., bilinear interpolation Ikehata (2022), pixel shuffle Ikehata (2023)) either smooth fine details or disrupt high-frequency semantics Odena et al. (2016), leading to degraded normal quality, especially in regions with complex variation (see Fig. 1 left).

In this paper, we propose **LINO UniPS**, a ViT-based framework that explicitly decouples illumination from surface cues while preserving geometric detail. First, we introduce learnable *Light Register Tokens* to capture illumination information. To handle diverse lights, we design three distinct types of tokens, each dedicated to a specific light source: point, directional lights, and environment (Env) lights. Our key innovation is an explicit *Light Alignment* supervision, which guides each register token to learn the characteristics of its corresponding illumination type. Second, we introduce an *Interleaved Attention Block* to process the image features alongside these specialized registers. This block employs a global cross-image attention mechanism that simultaneously attends to all tokens from all lighting conditions. The synergy between the Light Register Tokens and the Interleaved Attention Block enables the decoupling of lighting and normal cues, allowing our encoder to yield a **unified feature representation**. In parallel, we introduce two components to ensure the preservation of fine-scale geometry: a *Wavelet-based Dual-branch Architecture*, which introduces the wavelet domain to help the model better attend to high-frequency details, and a *Normal-gradient Perception Loss* that more heavily penalizes errors in high-frequency regions during training, thereby further refining local predictions. We further construct *PS-Verse*, a new dataset graded by surface complexity and lighting diversity, which enhances robustness and generalization under challenging real-world lighting. Together, these designs enable state-of-the-art accuracy, robustness, and generalization across synthetic and real benchmarks.

Overall, our contributions can be summarized as the following three points:

1. We introduce Light Register Tokens and an Interleaved Attention Block to explicitly decouple illumination from normal features, thereby yielding a unified feature representation.

2. We adopt a Wavelet-based Dual-branch Architecture and a Normal-gradient Perception Loss, which together substantially improve the reconstruction of fine-grained geometric details.

3. We build PS-Verse, a synthetic dataset graded by surface complexity and lighting diversity, and demonstrate superior accuracy and generalization through curriculum training.

# 2 RELATED WORK

**Calibrated Photometric Stereo:** Calibrated PS Woodham (1980) deduces surface normals by assuming meticulously pre-calibrated illumination, often relying on precise light source parameters. This stringent requirement limits the applicability of early methods. Initial methods successfully addressed Lambertian surfaces under such known lighting Woodham (1980). Subsequent research

extended capabilities to handle complex non-Lambertian reflectance, including varied BRDFs and specular highlights Grossberg & Nayar (2003); Chandraker et al. (2005); Goldman et al. (2005; 2010); Ikehata et al. (2012), albeit always presupposing perfectly known illumination conditions. More recently, deep learning techniques Jung et al. (2015); Santo et al. (2017); Chen et al. (2018); Ju et al. (2021) have significantly enhanced the ability to model intricate materials within this calibrated framework. However, the reliance of Calibrated PS on precise light source parameters severely restricts its applications to controlled laboratory environments, thereby limiting its real-world utility.

**Uncalibrated Photometric Stereo:** To alleviate photometric stereo's dependence on pre-calibrated light sources, the task of Uncalibrated Photometric Stereo (Uncalibrated PS) was introduced. Uncalibrated PS methods Hayakawa (1994); Belhumeur & Kriegman (1996); Mallick et al. (2005); Chen et al. (2020); Taniai & Maehara (2018); Kaya et al. (2021); Lichy et al. (2022) commonly adopt a two-stage pipeline: an initial stage where a network module estimates the unknown lighting parameters, followed by a second stage where these estimates are utilized as known inputs, like conventional Calibrated PS frameworks. While such Uncalibrated PS techniques have demonstrated success in handling more complex scenarios Lichy et al. (2022), their application under unconstrained natural or ambient illumination encounters significant challenges. These challenges primarily stem from the inherent difficulty in accurately modeling the physics of such arbitrary and often intricate lighting environments.

**Universal Photometric Stereo:** The Universal PS task, as recently formulated by Ikehata in Ikehata (2022), leverages a purely data-driven approach to solve the photometric stereo problem. This task enables operation under arbitrary and unknown lighting environments, critically obviating the need for complex predefined assumptions regarding the illumination. Building on this foundation, methods such as SDM UniPS Ikehata (2023) further advanced Universal PS by employing an encoder to extract features from multi-illumination images; these features are then fused to create a global lighting context, which a subsequent decoder utilizes in conjunction with pixel-level information to estimate surface normals. Uni MS-PS Hardy et al. (2024) has explored multi-scale strategies to address the Universal PS problem; however, such approaches often fall short of reconstructing surface normals that are simultaneously detailed and accurate. Our analysis of prior methods like UniPS and SDM UniPS indicates that an encoder's ability to extract features with greater consistency significantly facilitates the decoder's task of producing higher-fidelity normal maps. Motivated by this observation, we propose our method, LINO UniPS, which is designed to learn a superior *Unified Feature Representation* for Universal Photometric Stereo. Furthermore, we adopt a Wavelet-based Dual-branch Architecture and a Normal-gradient Perception Loss to address the challenge of detailed reconstruction in high-frequency regions.

## 3 METHOD

The task of Universal PS is to accurately recover the surface normal map $N \in \mathbb{R}^{H \times W \times 3}$ of an object or scene from a set of $F$ images $\{I_f\}_{f=1}^{F}$, where $I_f \in \mathbb{R}^{H \times W \times 3}$, captured under a single viewpoint but multiple unknown lighting conditions $\{\mathbf{L}_f\}_{f=1}^{F}$ Ikehata (2022).

As illustrated in Fig. 2, our method processes multi-light images through an encoder-decoder architecture. Within the encoder, the pipeline consists of three main stages: 1) First, the input images undergo processing by the (I) Wavelet Feature Extractor. This module simultaneously generates downsample components and low- and high-frequency wavelet components via wavelet decomposition Daubechies (1990); Finder et al. (2024), and then converts both components into independent token sequences through the backbone Dosovitskiy et al. (2021). 2) These sequences then enter the (II) Light Registered Attention Module, where they are prepended with specialized Light Register Tokens (Point, Direction, Env) to aggregate global illumination, before being processed by cascaded Interleaved Attention Blocks. 3) Next, these tokens, excluding the Light Register Tokens, are fed into the (III) Wavelet Aggregator. Here, features from the downsample and wavelet branches are fused to produce the **unified feature representation**, which is finally passed to a decoder for normal prediction.

The training is guided by two supervisory signals. First, a Light Alignment loss ensures that the register tokens capture the characteristics of their corresponding illumination sources. Second, a Normal Gradient Perception Loss enhances the model's ability to reconstruct fine-grained geometric details. Furthermore, this section only provides a high-level overview; please refer to appendix A1.1 for details.

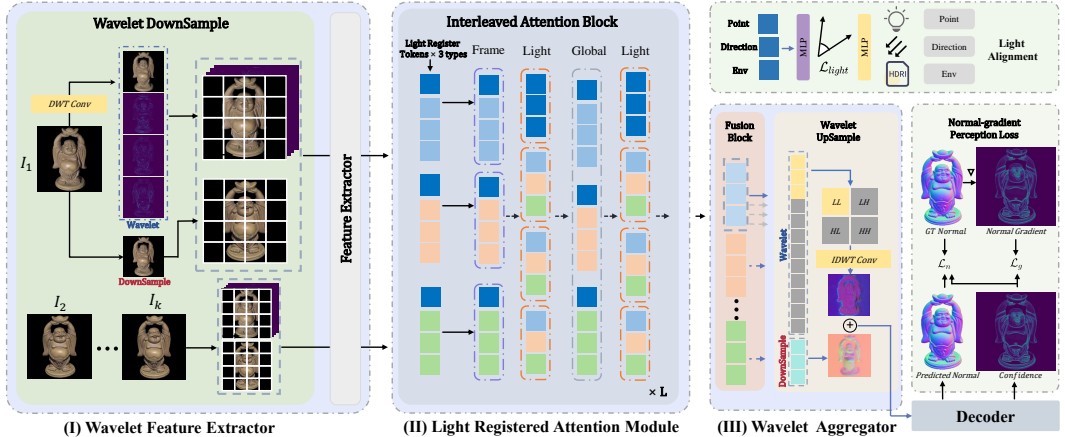

Figure 2: Overview of the LINO UniPS architecture. The encoder includes (I) Wavelet Feature Extractor, (II) Light Registered Attention Module and (III) Wavelet Aggregator, which together fuse wavelet and downsample domain features for obtaining unified feature.

## 3.1 LINO-UNIPS

Despite recent progress in this field, prior methods still suffer from two primary challenges.

*Challenge 1: The encoder fails to effectively decouple illumination and normal information.*

This failure forces the decoder to perform this disentanglement in addition to its primary prediction task. This creates a paradoxical workflow, as the more powerful encoder partially offloads the most difficult disentanglement to the weaker decoder. Motivated by this, our approach empowers the encoder to generate consistent, disentangled normal features. This simplifies the decoder's role to that of a simple refiner, easing the learning process. To achieve this, our method introduces two core components for this purpose: Light Register Tokens and Interleaved Attention Block.

**Light Register Tokens:** Inspired by the register mechanism in DINO Darcet et al. (2024), we introduce Light Register Tokens to aggregate global illumination information. To account for varied illumination types, we design distinct register tokens for point, direction, and environment (Env) lights. Our key innovation, and a crucial departure from DINO's unsupervised registers, is the introduction of an explicit Light Alignment supervision.

Drawing on the feature alignment methods used in VAVAE Yao et al. (2025) and REPA Yu et al. (2025) to accelerate generative model training, we encode the three types of Light Register Tokens to align with information about the light sources in the training dataset, including the point, direction and environment lights, (see Fig. 2, upper right corner), and supervise each of them separately with a cosine similarity loss. We summarize the form of the loss functions as follows:

$$\mathcal{L}_{\text{light}} = \lambda_1 \mathcal{L}_{\text{env}} + \lambda_2 \mathcal{L}_{\text{point}} + \lambda_3 \mathcal{L}_{\text{direction}} \tag{1}$$

where $\lambda_1$, $\lambda_2$, and $\lambda_3$ are hyperparameters.The settings for these hyperparameters and the detailed formulations of the $\mathcal{L}_{\text{env}}$, $\mathcal{L}_{\text{point}}$, and $\mathcal{L}_{\text{direction}}$ are provided in appendix A1.1.7 and appendix A1.1.3.

Through Light Alignment supervision, these tokens learn to capture and encode their respective illumination information, thereby achieving decoupling of lighting from normal features. This is demonstrated by the phenomena observed in Fig. 3. The attention maps show that the register tokens have learned specialized focus. Specifically, the Point register tokens exhibit a sparse and sharp attentional focus, concentrating on high-intensity regions, which strongly aligns with the high-frequency characteristics of point light sources that produce distinct specular highlights. The Direction and Env register tokens attend to broad, spatially-diffuse regions across the object, aligning perfectly with the expected low-frequency characteristics of directional and environmental illumination that govern global shading and softer shadows. This clear division of labor confirms the effectiveness of our register tokens in disentangling the different physical components of the illumination.

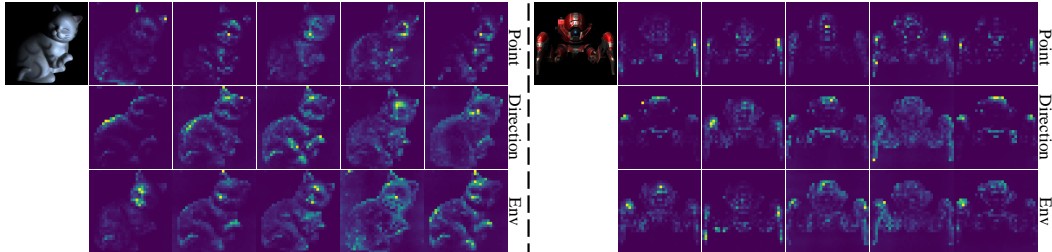

Figure 3: Attention maps for our different Light Register Tokens, derived from the encoder's final-layer feature maps for both real (left) and synthetic (right) data. These maps visually confirm that the tokens learn to focus on distinct distributions of illumination information.

**Interleaved Attention Block:** Previous methods like UniPS and SDM UniPS rely solely on Frame Attention (Frame) Dosovitskiy et al. (2021) and Light Axis Attention (Light) Ikehata (2023). While these mechanisms facilitate local information flow, they are insufficient for effectively decoupling lighting variations and normal features. Inspired by VGGT Wang et al. (2025), our Interleaved Attention block introduces a more powerful global cross-image attention mechanism (Global) that attends to all input image tokens simultaneously. Specifically, our block applies four attention layers in an interleaved sequence: Frame → Light → Global → Light (see Fig. 2, middle).

The key advantage of this design is its ability to aggregate and fuse information across multiple hierarchical levels. Light-axis attention operates at the patch level, frame attention captures intra-image context, and our global attention integrates information at the cross-image level. This multi-level awareness enables the model to build a holistic understanding of the global illumination from local to global scales, thereby achieving a better decoupling from the intrinsic normal features.

*Challenge 2: The model fails to preserve high-frequency geometric details.*

Accurate normal reconstruction is highly dependent on preserving high-frequency geometric details, yet current feature extraction techniques often discard them. Although prior methods recognize this issue, their solutions are inadequate. UniPS Ikehata (2022) extracts features from downsampled images, which inherently introduces blurring. While SDM UniPS Ikehata (2023) attempts to mitigate this with a "split-and-merge" operation, this process can disrupt high-frequency semantics Odena et al. (2016). To overcome these limitations, we propose a two-pronged approach: a Wavelet-based Dual-branch Architecture to preserve information during feature extraction, and a Normal Gradient Perception Loss to guide the model's focus towards these fine details during training.

**Wavelet-based Dual-branch Architecture:** To mitigate information loss during downsampling, our dual-branch uses the discrete wavelet transform Daubechies (1990). This allows us to decompose multi-light images into their high- and low-frequency components. At the same time, to retain global image-domain semantic information, we maintain a parallel branch that performs downsampling. During feature upsampling, an inverse wavelet transform reconstructs features from the wavelet domain, ensuring detail preservation throughout the network.

**Normal Gradient Perception Loss:** To further guide the network's focus towards complex geometric and richly textured regions, we introduce a Normal Gradient Perception Loss ($\mathcal{L}_\mathrm{n}$). Instead of treating all pixels equally, this loss uses the predicted normal gradient ($\tilde{G}$) to generate a confidence map ($C = e^{\tilde{G}}$) that amplifies the error signal in high-frequency areas. The loss is a weighted sum of this confidence-weighted reconstruction error and a gradient supervision term:

$$\mathcal{L}_\mathrm{n} = \lambda_4 \sum (N - \tilde{N})^2 \odot C + \lambda_5 \sum (\tilde{G} - G)^2, \tag{2}$$

Here, the first term penalizes the difference between the predicted normal ($\tilde{N}$) and ground truth ($N$), weighted by the confidence map $C$. The second term directly supervises the predicted gradient ($\tilde{G}$) against the ground truth gradient ($G = \nabla N$). The coefficients $\lambda_4$ and $\lambda_5$ balance these two objectives. This design makes the network explicitly sensitive to fine surface details, significantly improving reconstruction quality in challenging regions.

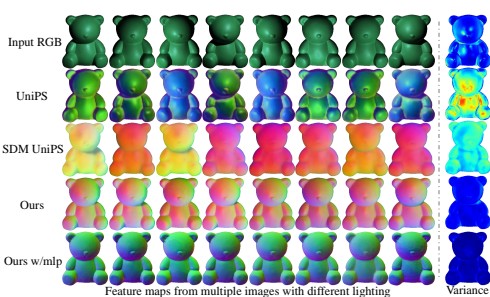
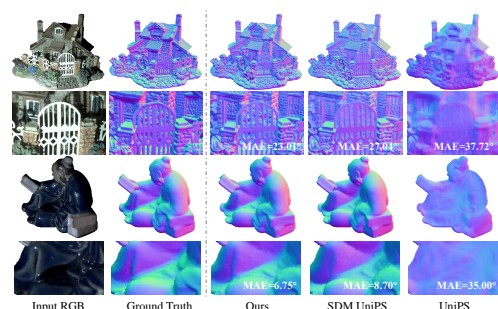

Figure 4: Features from different methods' encoders; rightmost column is variance.

Figure 5: Results of object inference with masks in Luces and DiLiGenT. Using 16 input images.

Table 1: Comparison of PS-Verse with Other Photometric Stereo Datasets. The terms 'Intrinsic', 'Normal', and 'Light' indicate the availability of material parameters (albedo, etc.), the use of normal maps for rendering detail, and the presence of light type annotations. 'Complexity' quantifies the average magnitude of surface normal gradients, where higher values denote greater geometric intricacy. 'Shapes', 'Env', and 'Scenes' are the respective counts of 3D models, HDRI environment maps, and rendered scenes.

| Dataset | Intrinsic | Normal | Light | Complexity | # Shapes | # Env | # Scenes |
|---|---|---|---|---|---|---|---|
| CyclesPS-Train Ikehata (2018) | ✗ | ✗ | ✓ | 4.9 | 15 | 0 | 45 |
| PS-Wild Ikehata (2022) | ✗ | ✗ | ✗ | 3.5 | 410 | 31 | 10,099 |
| PS-Mix Ikehata (2023) | ✓ | ✗ | ✗ | 11.5 | 410 | 31 | 34,927 |
| PS-Uni MS-PS Hardy et al. (2024) | ✗ | ✗ | ✓ | 8.6 | 11,000 | 1100 | **100,000** |
| PS-Verse | ✓ | ✓ | ✓ | **26.7** | **17,805** | **2423** | **100,000** |

## 3.2 PS-VERSE DATASET

Prior large-scale synthetic datasets, such as PS-Wild Ikehata (2022) and PS-Mix Ikehata (2023), have advanced the field by enabling data-driven Universal PS. However, they remain limited by either overly simplistic lighting (e.g., lacking high-frequency point sources) or geometrically simple objects that fail to produce complex lighting variations and self-shadowing effects.

Leveraging publicly available large-scale 3D asset datasets Deitke et al. (2022; 2023); Vecchio & Deschaintre (2024), we propose the PS-Verse dataset (see Tab. 1 for a detailed comparison with other Photometric Stereo datasets). To increase the lighting complexity in rendered scenes, we carefully select 17,805 textured 3D models with UV coordinates and PBR materials from the Objaverse dataset Deitke et al. (2022), which contains nearly 800,000 models. Following Dora Chen et al. (2024), which quantifies geometric complexity based on the density of sharp edges, we categorize the objects into four complexity levels. We then construct scenes by recursively selecting 4 to 6 objects from each level, and classify scenes into four corresponding complexity tiers.

Regarding materials, textures are randomly sampled from the MatSynth large-scale PBR material database Vecchio & Deschaintre (2024), consisting of 806 metallic, 1,226 specular, and 3,321 diffuse material groups. To balance the distribution of specular and diffuse lighting effects, we assign object materials according to the ratio of 1:4:2.5:2.5 across original object textures, diffuse, specular, and metallic materials. To more realistically simulate fine surface detail interactions with light, we introduce normal mapping for objects for the first time in such data, defining this as a fifth complexity level that injects rich high-frequency lighting details.

In terms of lighting setup, we use uniform environment light to simulate non-dark ambient conditions. Detailed lighting configurations are provided in the appendix A1.3.3. Overall, the PS-Verse dataset consists of 100,000 scenes generated with Blender Foundation. Each scene includes two rendering outputs: with and without normal mapping. We render 20 images at a resolution of 512 per scene.

In addition to providing ground truth for surface normals, the PS-Verse dataset also offers ground truth for albedo, roughness, and metallic, to support PBR prediction tasks. The PBR prediction results and visual showcase of the PS-Verse dataset can be found in the appendix A1.2 and A1.3.

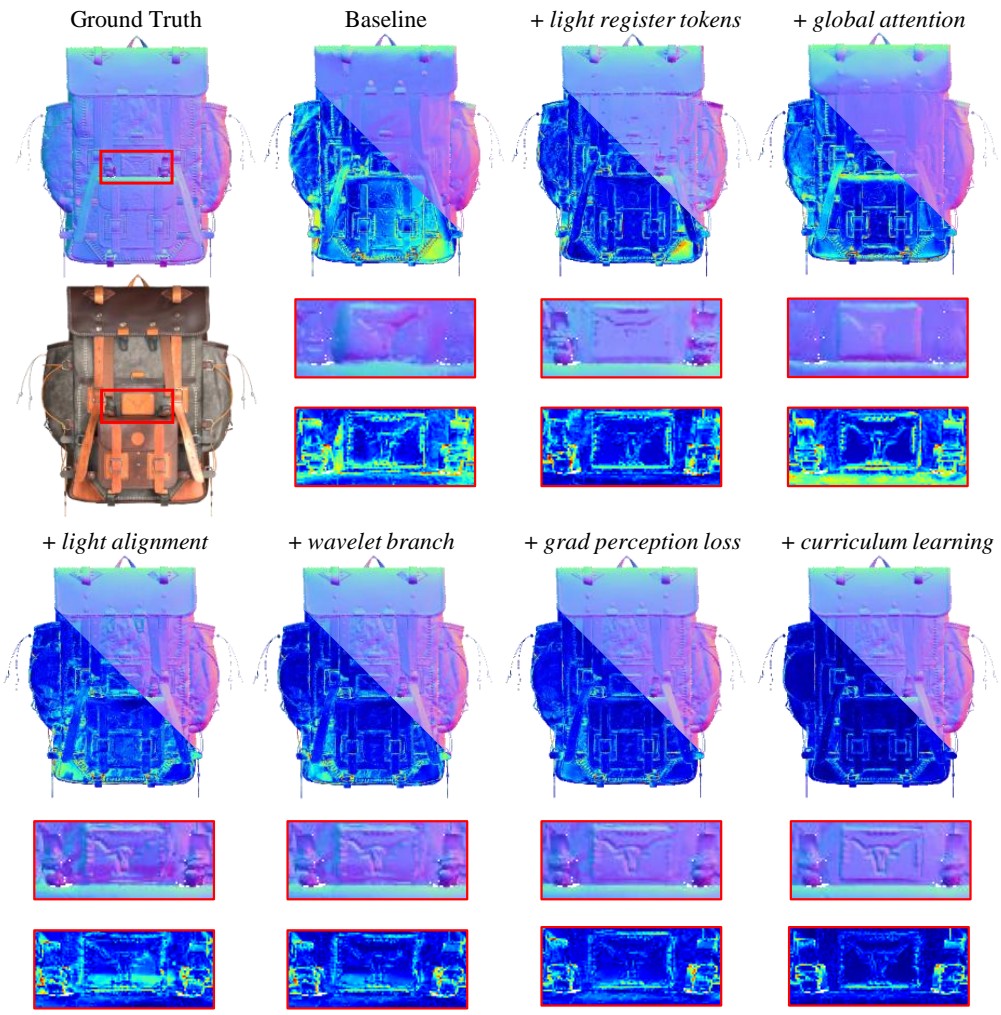

Figure 6: Qualitative results for the main ablation study. The visual comparisons are shown on samples from our PS-Verse Testdata. We recommend zooming in to observe fine-grained details.

## 4 EXPERIMENTS

**Implementation Details:** To effectively enhance our method's capability for reconstructing fine-grained surface normals, we employ a curriculum learning strategy that progresses from low to high geometric complexity. We start training on PS-Verse Level 1 data, adding higher levels every 10 epochs up to Level 4, for about 150 epochs. Then, we finetune on Level 5 data with ground truth normals until 200 epochs to improve surface detail reconstruction. We use AdamW Optimizer with $1e^{-4}$ initial learning rate, $0.05$ weight decay and a step decay of $0.8$ every ten epochs. Input images per batch vary randomly from 3 to 6. The total loss $\mathcal{L}$ combines multiple components:

$$\mathcal{L} = \mathcal{L}_{\text{light}} + \mathcal{L}_{\text{n}} \tag{3}$$

Training runs on 2 NVIDIA H100 GPUs for roughly 3 days. Inference takes around 1.5 seconds for 16 input images at 512×512 on H100.

**Evaluation Metric:** We evaluate our results using three metrics. For surface normal accuracy, we measure the Mean Angular Error (MAE, in degrees, ↓). To assess feature similarity, we utilize Cosine Similarity (CSIM ↑) on normalized features and the Structural Similarity (SSIM ↑) Wang et al. (2004) on their 3D PCA projections. For both metrics, the final score is obtained by averaging the similarity values computed between all possible feature pairs. For instance, given a set of 6 features, we calculate all $C_6^2$ pairwise values to derive this mean.

**Evaluation Dataset:** For evaluation, comparison experiments and ablation studies, we employ two public benchmarks, DiLiGenT Shi et al. (2018) and Luces Logothetis et al. (2022), and our synthetic PS-Verse Testdata. This set contains 441 scenes held out from the training data, with each scene featuring a single object to allow for clearer qualitative evaluation. To verify the model's generalization capabilities, we test on real-world images from SDM UniPS Ikehata (2023).

## 4.1 ABLATION STUDY AND COMPARISON EXPERIMENT

**Ablation:** The main ablation quantitative results are presented in Tab. 2, and the qualitative results are in Fig. 6. **First**, to demonstrate the superiority of our PS-Verse dataset, we retrain the Uni MS-PS Hardy et al. (2024) on both the PS-Mix Ikehata (2023) and our PS-Verse. We then compare these models against the officially released Uni MS-PS, which was trained on its native data (PS-Uni MS-PS). The results show that the model trained on PS-Verse yields substantially better feature similarity scores (CSIM and SSIM) and a lower MAE for normal reconstruction, which strongly indicates that PS-Verse is more effective for training high-performance models.

**Second**, we conduct ablations of the various modules within our LINO UniPS method, starting from a baseline model where all our proposed enhancements are removed. Our initial finding is that by merely incorporating unsupervised Light Register Tokens, both the feature similarity metrics and the normal reconstruction results show an improvement. This suggests

Table 2: Main ablation with 20 multi-lights input images. The evaluation metrics were measured on our PS-Verse Testdata.

| Method | Dataset | CSIM↑ | SSIM↑ | Avg. MAE↓ |
|---|---|---|---|---|
| Uni MS-PS | PS-Uni MS-PS | 0.72 | 0.70 | 9.02 |
| Uni MS-PS | PS-Mix | 0.63 | 0.66 | 10.02 |
| Uni MS-PS | PS-Verse | 0.75 | 0.73 | 7.82 |
| Baseline | PS-Verse | 0.71 | 0.69 | 8.73 |
| + *light register tokens* | PS-Verse | 0.74 | 0.73 | 8.13 (0.60 ↓) |
| + *global attention* | PS-Verse | 0.80 | 0.78 | 6.44 (2.29 ↓) |
| + *light alignment* | PS-Verse | 0.86 | 0.82 | 5.58 (3.15 ↓) |
| + *wavelet branch* | PS-Verse | 0.85 | 0.82 | 5.15 (3.58 ↓) |
| + *grad perception loss* | PS-Verse | 0.86 | 0.83 | 4.84 (3.89 ↓) |
| + *curriculum learning* | PS-Verse | **0.88** | **0.86** | **4.51** (4.22 ↓) |

that these additional tokens can often capture global lighting information even without direct supervision, thereby aiding the disentanglement of normals from illumination. Progressively incorporating our global cross-image attention mechanism to our interleaved Attention Block and light alignment supervision leads to more significant improvements in both feature similarity and normal reconstruction performance. This further validates our central conclusion: that more effectively decoupling lighting and normal features enhances the quality of normal reconstruction. Next, we integrate the wavelet branch to form our Wavelet-based Dual-branch Architecture and the Normal-gradient Perception Loss for extracting fine-grained context. While these two modules have a modest impact on feature similarity metrics, they substantially improve performance on data with complex geometries as intended. Finally, from the last row, adopting curriculum learning boosts all metrics, improving feature similarity while also reducing MAE, which confirms the effectiveness of this training strategy.

Table 3: Architectural comparison on the PS-Mix dataset Ikehata (2023). The consistency of the encoder features is measured by Cosine Similarity (CSIM) and Structural Similarity (SSIM), while the final normal reconstruction accuracy is evaluated by Mean Angular Error (MAE).

| Method | CSIM↑ | SSIM↑ | MAE↓ | |
|---|---|---|---|---|
| | | | DiLiGenT | Luces |
| SDM UniPS Ikehata (2023) | 0.84 | 0.72 | 5.80 | 13.50 |
| Uni MS-PS Hardy et al. (2024) | 0.82 | 0.76 | 5.75 | 13.71 |
| Ours | **0.90** | **0.91** | **5.60** | **12.70** |

**Comparison 1:** To isolate the architectural advantages of LINO UniPS from the effects of training data, we retrain it alongside Uni MS-PS Hardy et al. (2024) on the PS-Mix dataset Ikehata (2023) to convergence ( 100 epochs). As presented in Tab. 3, our LINO UniPS surpasses SDM UniPS and Uni MS-PS across both evaluated public datasets. Notably, LINO UniPS yields higher CSIM and SSIM scores, quantitatively demonstrating that features extracted by our encoder exhibit greater similarity compared to those from SDM UniPS Ikehata (2023) and Uni MS-PS. It is noteworthy that our LINO UniPS utilizes a decoder architecture identical to that of SDM UniPS Ikehata (2023). This architectural commonality strongly suggests that the observed performance improvements are primarily attributable to our encoder's enhanced capability to more effectively decouple illumination from geometry. Such effective

Table 4: Evaluation on DiLiGenT Shi et al. (2018). Uses all 96 images unless otherwise noted (K). Normal reconstruction accuracy is evaluated by Mean Angular Error (MAE).

| Method | Ball | Bear | Buddha | Cat | Cow | Goblet | Harvest | Pot1 | Pot2 | Reading | Avg. MAE |
|---|---|---|---|---|---|---|---|---|---|---|---|
| UniPS Ikehata (2022) | 4.90 | 9.10 | 19.40 | 13.00 | 11.60 | 24.20 | 25.20 | 10.80 | 9.90 | 18.80 | 14.70 |
| SDM UniPS Ikehata (2023) | 1.50 | 3.60 | 7.50 | 5.40 | 4.50 | 8.50 | 10.20 | 4.70 | **4.10** | 8.20 | 5.80 |
| Uni MS-PS Hardy et al. (2024) | 1.92 | 3.14 | 6.16 | 3.60 | 4.04 | 6.35 | 8.84 | 4.08 | 4.88 | 7.09 | 5.01 |
| Ours w/ mlp | **1.21** | 3.62 | 7.36 | 4.83 | 4.94 | 6.11 | 10.71 | 5.37 | 5.23 | 7.54 | 5.69 |
| Ours | 1.74 | **2.64** | **6.12** | **3.38** | **3.99** | **5.17** | **8.58** | **4.07** | 4.14 | **6.67** | **4.65** |
| Ours(K=32) | 1.75 | 2.66 | 6.26 | 3.49 | 4.06 | 5.25 | 8.71 | 4.12 | 4.26 | 6.75 | 4.73 |
| Ours(K=16) | 1.92 | 2.74 | 6.40 | 3.51 | 4.25 | 5.41 | 8.81 | 4.14 | 4.45 | 7.12 | 4.88 |

Table 5: Evaluation on Luces Mecca et al. (2021). Uses all 52 images unless otherwise noted (K). Normal reconstruction accuracy is evaluated by Mean Angular Error (MAE).

| Method | Ball | Bell | Bowl | Buddha | Bunny | Cup | Die | Hippo | House | Jar | Owl | Queen | Squirrel | Tool | Avg. MAE |
|---|---|---|---|---|---|---|---|---|---|---|---|---|---|---|---|
| UniPS Ikehata (2022) | 11.01 | 24.12 | 23.84 | 27.90 | 23.51 | 28.64 | 16.24 | 21.41 | 35.93 | 14.53 | 32.87 | 28.36 | 25.36 | 19.03 | 23.77 |
| SDM UniPS Ikehata (2023) | 13.30 | 12.76 | 8.44 | 18.58 | 8.53 | 19.67 | 7.25 | 8.86 | 26.07 | 8.30 | 12.67 | 15.97 | 16.01 | 12.54 | 13.50 |
| Uni MS-PS Hardy et al. (2024) | 10.20 | 10.52 | 6.98 | 12.83 | 9.60 | 13.68 | 6.19 | 8.33 | 25.29 | 6.30 | 11.47 | 12.45 | 11.36 | 11.79 | 11.21 |
| Ours w/ mlp | **9.09** | 12.00 | 10.09 | 16.63 | 9.87 | 15.97 | 6.86 | 9.44 | 25.37 | 7.65 | 11.77 | 13.62 | 16.57 | 13.22 | 12.62 |
| Ours | 10.16 | **8.78** | **6.96** | **12.67** | **6.09** | **8.15** | **6.16** | **5.99** | **22.91** | **6.24** | **9.58** | **9.84** | **10.25** | **8.25** | **9.43** |
| Ours (K=15) | 10.27 | 8.80 | 9.01 | 14.05 | 6.40 | 8.42 | 6.87 | 6.04 | 23.60 | 6.89 | 10.48 | 9.93 | 10.29 | 8.29 | 9.94 |

decoupling fosters stronger normal consistency within the learned features, consequently boosting the accuracy and capability of the final normal reconstruction.

**Comparison 2:** We present a comprehensive analysis of the parameter count in Tab. 6. To investigate the impact of model size, we finetune two smaller variants on PS-Verse, Ours-S1 (73.2M) and Ours-S2 (60.4M), by reducing the number of layers in Feature Extractor and Interleaved Attention Blocks. These models are designed to have parameter counts comparable to Uni MS-PS (75.5M) and SDM UniPS (59.9M). The results clearly show that our models consistently outperform their counterparts at similar parameter scales, confirming that the superiority of our method stems from its architectural design, not merely from a larger parameter count.

Furthermore, the table compares inference times. On both high-resolution (4000×4000) and standard-resolution (512×612) images, our method runs slightly faster than SDM UniPS and is substantially more efficient than Uni MS-PS, which requires approximately **35 times more computation time**. This highlights the efficiency of our approach, particularly for high-resolution processing.

## 4.2 QUANTITATIVE RESULTS

As observed in Tab. 4, on the DiLiGenT Shi et al. (2018), our method achieves a new SOTA performance with an average MAE of 4.65°, outperforming the previous best method. Similar results can also be seen in Tab. 5. On Luces Mecca et al. (2021), a dataset featuring high-frequency information, our method obtains an average MAE of 9.43°. This represents a substantial improvement over the prior best performance (11.21°). Some qualitative results are presented in Fig. 5.

To further underscore the efficacy of our encoder, we conduct an additional experiment. In this setup, we replace the standard decoder in our LINO UniPS with a simpler Multi-Layer Perceptron (MLP) and then finetune this variant until convergence. Our findings are twofold: First, the consistency of the extracted features is further enhanced, as detailed in Fig. 4. Second, although the reconstruction performance sees a slight degradation compared to LINO UniPS with its original, more sophisticated decoder, this MLP-decoder variant (w/mlp) still outperforms SDM UniPS. These results align with our hypothesis that the superior performance of LINO UniPS is primarily driven by its encoder's advanced capability to generate highly consistent and well-disentangled features.

Moreover, as detailed in Tab. 4 and Tab. 5, our method achieves SOTA on all scenes except for the 'Ball' object in both benchmarks, demonstrating the general superiority of our approach. Interestingly, on the 'Ball' scenes, the best performance is achieved by our MLP-decoder variant. We hypothesize that this is because a simpler MLP decoder is better suited for recovering geometrically simple primitives like the 'Ball'.

Table 6: Ablation study on parameter count. Inference time is measured on 16 images (K=16). The $4000 \times 4000$ and $512 \times 612$ resolution images are sourced from the SDM UniPS real data Ikehata (2023) and the DiLiGenT Shi et al. (2018).

| Method | Params (M) | Inference Time (s) | | MAE (°)↓ | |
|---|---|---|---|---|---|
| | | $4000 \times 4000$ | $512 \times 612$ | DiLiGenT | LUCES |
| Ours | 84.2 | 85.1 | 1.7 | 4.65 | 9.43 |
| Ours-S1 | 73.2 | 82.9 | 1.5 | 4.83 | 10.05 |
| Ours-S2 | 60.4 | 81.0 | 1.5 | 4.95 | 10.89 |
| SDM UniPS | 59.9 | 92.7 | 1.8 | 5.83 | 13.52 |
| Uni MS-PS | 75.5 | 3012.2 | 35.3 | 5.01 | 11.21 |

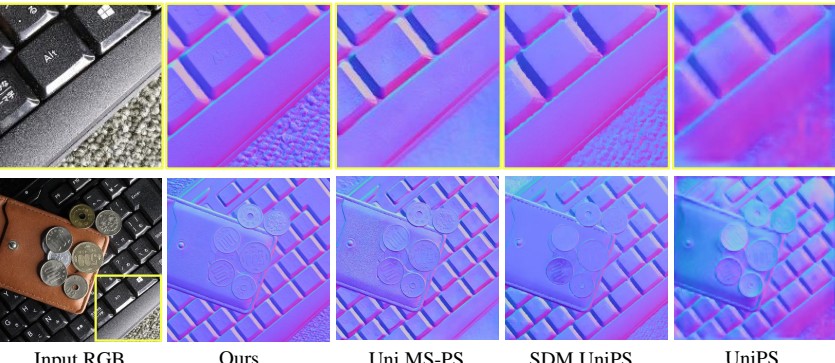

| Input RGB | Ours | Uni MS-PS | SDM UniPS | UniPS |
|---|---|---|---|---|

Figure 7: Qualitative comparison for the high-resolution (4K) 'Coins and keyboard' scene from SDM UniPS Ikehata (2023) with 8 input images. Our method recovers more intricate details than the baseline.

## 4.3 QUALITATIVE RESULTS

Fig. 1 and Fig. 7 showcase our method's ability to reconstruct highly detailed and accurate surface normals for real-world objects and scenes, highlighting its strong generalization capabilities. As shown in Fig. 1, when processing high-resolution images, a close inspection of detail-rich regions like the rabbit's ears and abdomen reveals a clear distinction. While UniPS and SDM UniPS produce over-smoothed results and fail to capture the intricate surface geometry, our LINO UniPS successfully reconstructs fine-grained details in these challenging areas. Fig. 7 further demonstrates our method's performance on more complex, high-resolution (4K) real-world scenes. Our method consistently produces more physically plausible and detailed results. For instance, a key differentiator is revealed in the background textile, as shown in the magnified view of the top row. Here, Uni MS-PS, SDM UniPS and UniPS fail to reconstruct the complex texture of the tablecloth, whereas our method demonstrates stronger generalization by accurately recovering its intricate fabric pattern. Additional reconstruction results on real-world scenarios can be found in appendix A1.5.2.

## 5 CONCLUSION

In this paper, we propose LINO UniPS, a novel framework for Universal Photometric Stereo that addresses two core challenges. The first is the failure to disentangle illumination-invariant surface normals from spatially-varying lighting. To this end, we employ Light Register Tokens with an explicit light alignment and an Interleaved Attention Block with a global cross-image attention mechanism. These components work in concert to capture the global lighting context and enable better separation of lighting and normal features. The second challenge is the loss of fine-grained detail. To address this, we integrate a wavelet branch to form a Wavelet-based Dual-branch Architecture and introduce a Normal-gradient Perception Loss, which heightens the model's sensitivity to intricate geometry. Finally, we contribute a complex and large-scale photometric stereo dataset, hoping to provide valuable reference for future research.

## ETHICS STATEMENT.

This work does not involve human subjects, personally identifiable information, or sensitive data. The datasets used in this study are publicly available and widely adopted in the machine learning community. All experiments were conducted using standard computational resources without environmental or societal harm. The methodology does not introduce discriminatory biases, and the model's potential applications are aligned with responsible AI principles. The authors have reviewed the ICLR Code of Ethics and confirm that this submission adheres to its guidelines.

## REPRODUCIBILITY STATEMENT.

To support reproducibility, we provide a complete description of our model architecture, training procedures, hyperparameters, and evaluation protocols in the main paper. Additional implementation details are included in the appendix. We have strived to document all necessary components with sufficient clarity to enable independent replication of our results.

## ACKNOWLEDGEMENTS.

This work was partially supported by a grant from the NSFC/RGC Collaborative Research Scheme sponsored by the Research Grants Council of the Hong Kong Special Administrative Region, China and National Natural Science Foundation of China (Project No. CRS_HKUST605/25).

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

# A1  APPENDIX

This supplementary document offers further technical details, demonstrations of the datasets employed, additional insights, and comprehensive results pertaining to our LINO UniPS.

## A1.1  NETWORK ARCHITECTURE DETAILS

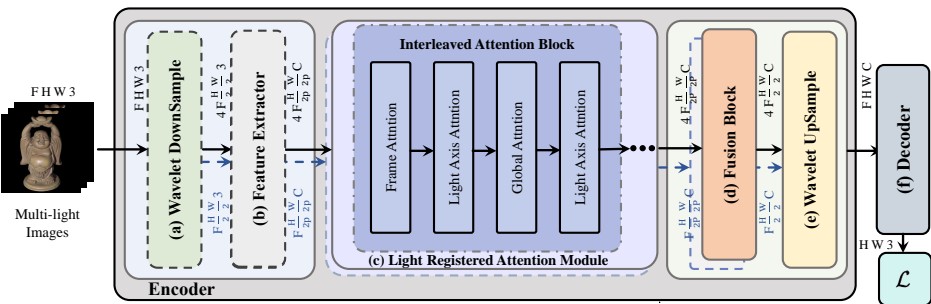

Figure A1: An overview of our network architecture, illustrating the corresponding tensor shape transformations. Black solid arrows denote the forward pass of the downsample branch, while purple dashed arrows denote the forward pass of the wavelet branch.

To provide readers with a more in-depth understanding of our LINO UniPS network architecture, in the following, we detail the architecture of our LINO UniPS, breaking it down into several key modules: (a) Wavelet DownSample Module, (b) Feature Extractor, (c) Light Registered Attention Module, (d) Fusion Block, (e) Wavelet UpSample Module, (f) Decoder and the training loss. An overview of our network architecture is presented in Fig. A1. In the subsequent sections, we will provide a detailed account of the network's structural components and the specific transformations of tensor shapes as data progresses through the model.

### A1.1.1  WAVELET DOWNSAMPLE MODULE:

To enable our encoder to extract more fine-grained contexts, we first incorporate the wavelet transform Daubechies (1990), chosen for its ability to separate an image's high- and low-frequency components Finder et al. (2024); Huang et al. (2017) while concurrently mitigating losses typically incurred during downsampling Peng et al. (2024).

Initially, the input to our network is a batch of multi-light image sets, represented by a tensor $I \in \mathbb{R}^{B \times F \times H \times W \times 3}$. In this notation, $B$ denotes the batch size, $F$ is the number of images captured under different illumination conditions for each scene instance, $H$ and $W$ represent the spatial dimensions (height and width), and the final dimension 3 corresponds to the RGB color channels. To simplify the subsequent exposition, we will assume a batch size of $B = 1$ unless otherwise specified, effectively considering the processing of a single multi-illumination image set at a time. As part of the preprocessing, to ensure that image pixel values lie within a comparable range, each of the $F$ images within a given scene instance is normalized by a random scalar sampled uniformly between its maximum and mean values. Following this preprocessing, for the purpose of subsequent discussion (effectively assuming $B = 1$), we obtain a set of $F$ images $\{I_f\}_{f=1}^{F}$, where each $I_f \in \mathbb{R}^{H \times W \times 3}$.

Following SDM UniPS Ikehata (2023), for each pre-processed input image $I_f \in \mathbb{R}^{H \times W \times 3}$, we perform two separate transformations: naive downsampling to obtain $I_f^d \in \mathbb{R}^{\frac{H}{2} \times \frac{W}{2} \times 3}$ in the image domain, and a wavelet transform to yield its corresponding wavelet domain components $I_f^w \in \mathbb{R}^{4 \times \frac{H}{2} \times \frac{W}{2} \times 3}$, namely $I_f^{ll} \in \mathbb{R}^{\frac{H}{2} \times \frac{W}{2} \times 3}$, $I_f^{lh} \in \mathbb{R}^{\frac{H}{2} \times \frac{W}{2} \times 3}$, $I_f^{hl} \in \mathbb{R}^{\frac{H}{2} \times \frac{W}{2} \times 3}$, and $I_f^{hh} \in \mathbb{R}^{\frac{H}{2} \times \frac{W}{2} \times 3}$.

### A1.1.2  FEATURE EXTRACTOR:

Subsequently, both the downsample image representation $I_f^d \in \mathbb{R}^{\frac{H}{2} \times \frac{W}{2} \times 3}$ and the set of wavelet components $I_f^w$ (comprising $I_f^{ll}, I_f^{lh}, I_f^{hl}, I_f^{hh}$, each in $\mathbb{R}^{\frac{H}{2} \times \frac{W}{2} \times 3}$) are individually processed. First, each of these input components is partitioned into a sequence of patch-based tokens. These token sequences are then fed into our Feature Extractor backbone. This backbone is trained during our training procedure to extract rich visual representations from these diverse inputs. In our specific

implementation, we set the patch size to $P = 8$. Consequently, for each input component with spatial dimensions $H/2 \times W/2$, the resulting sequence length is $L = \frac{(H/2) \times (W/2)}{P^2}$ tokens. The feature embedding dimension is $D = 384$. Following processing by this backbone, we respectively obtain shallow visual feature representations $F_{s,f}^d \in \mathbb{R}^{L \times D}$ from the downsample image stream (derived from $I_f^d$) and $F_{s,f}^w \in \mathbb{R}^{4 \times L \times D}$ from the wavelet components stream (derived from $I_f^w$).

### A1.1.3 LIGHT REGISTERED ATTENTION MODULE:

To achieve a more effective decoupling of lighting and normal features that are subsequently processed by the decoder, we introduce our Light Registered Attention Module.

Firstly, we design the Light Register Tokens to improve the handling of global illumination. While lighting information predominantly exhibits global characteristics across multi-light inputs Ikehata (2018; 2022; 2023), traditional attention mechanisms in existing Universal PS methods often fail to fully leverage this distributed information. This deficiency can hinder effective illumination-geometry separation, motivating our specialized token-based strategy. Drawing inspiration from advancements like Darcet et al. (2024), and further considering the inherent illumination-dependency of the Universal PS task, we introduce these Light Register Tokens to explicitly capture and represent decoupled global lighting information within our framework.

To facilitate the perception of distinct lighting components, we additionally introduce three specialized Light Register Tokens: $x_{\text{env}} \in \mathbb{R}^{1 \times D}$, designated for perceiving environment light; $x_{\text{point}} \in \mathbb{R}^{1 \times D}$, tailored for point lights (which often contribute high-frequency illumination effects); and $x_{\text{direction}} \in \mathbb{R}^{1 \times D}$, for directional light (typically representing low-frequency illumination sources). Subsequently, this set of three specialized light tokens is prepended to the token sequences derived from $F_{s,f}^d$ (features from the downsample image stream) and $F_{s,f}^w$ (features from the wavelet components stream), respectively, leading to: $F_{s,f,r}^d \in \mathbb{R}^{L' \times D}$ and $F_{s,f,r}^w \in \mathbb{R}^{4 \times L' \times D}$, where $L' = L + 3$.

Then $F_{s,f,r}^d$ and $F_{s,f,r}^w$ are fed into our Interleaved Attention Block Wang et al. (2025) to enhance inter-intra feature communication. Specifically, our Interleaved Attention Block contains four attention layers, which can be represented as: Frame $\rightarrow$ Light $\rightarrow$ Global $\rightarrow$ Light.

Previous work has found that feature communication within the encoder is very important Ikehata (2022; 2023); Hardy et al. (2024), but their methods are often limited to patch-level local light-axis attention. Our Interleaved Attention Block, however, breaks such limitations. On the one hand, it incorporates Frame attention to enhance intra-image communication. On the other hand, we have added Global attention, a more comprehensive global operation, allowing inter-image features to also be extended from the patch level to the image level. Upon processing by these four cascaded Interleaved Attention Blocks (the number chosen to maintain a manageable parameter count, although more blocks could potentially be employed), we obtain the deep feature representations denoted as $F_{d,f,r}^d \in \mathbb{R}^{L' \times D}$ and $F_{d,f,r}^w \in \mathbb{R}^{4 \times L' \times D}$. It is worth noting that the attention operations inherent in these blocks do not alter the fundamental shapes of these tensor sequences.

To ensure the Light Register Tokens $x_{\text{env}}$, $x_{\text{point}}$, $x_{\text{direction}}$ effectively capture global illumination, as direct supervision of the decoded light map is challenging, we introduce a light-aware feature alignment strategy during training Yu et al. (2025); Yao et al. (2025); Song et al. (2025). Since we train LINO UniPS on our own rendered synthetic dataset PS-Verse, for every scene within this training dataset, we have access to its corresponding lighting information from the rendering process. This includes: the HDRI environment map, the positions, the distance to camera and intensities of point lights, and the positions, the distance to camera, intensities and areas of directional lights. Mathematically, these are denoted as $L_{\text{env}} \in \mathbb{R}^{H \times W \times 3}$, $L_{\text{point}} \in \mathbb{R}^{M_1 \times 5}$, and $L_{\text{direction}} \in \mathbb{R}^{M_2 \times 6}$, respectively. For $L_{\text{point}} \in \mathbb{R}^{M_1 \times 5}$, where $M_1$ is the number of point lights, its first three dimensions denote position, the fourth denotes the distance and the fifth denotes the intensity. For the directional light component $L_{\text{direction}} \in \mathbb{R}^{M_2 \times 6}$, where $M_2$ denotes the number of directional lights, its first three dimensions specify position, the fourth denotes the distance, the fifth denotes the directional light size, and the sixth indicates intensity. Subsequently, we encode the lighting components $L_{\text{env}}$, $L_{\text{point}}$, and $L_{\text{direction}}$ by projecting them into a $D$-dimensional feature space, thereby obtaining their respective representations $\mathbf{l}_{\text{env}}^h \in \mathbb{R}^D$, $\mathbf{l}_{\text{point}}^h \in \mathbb{R}^D$, and $\mathbf{l}_{\text{direction}}^h \in \mathbb{R}^D$. Similarly, the light tokens $x_{\text{env}}$, $x_{\text{point}}$, and $x_{\text{direction}}$, after being processed by the cascaded Interleaved Attention Blocks, are projected into the same $D$-dimensional feature space. This yields their respective high-dimensional

Table A1: Comparison of PBR material prediction performance. A dash (-) indicates the method does not provide the corresponding output.

| Method | Normal | Albedo | | Metallic | | Roughness | |
|---|---|---|---|---|---|---|---|
| | MAE↓ | PSNR↑ | SSIM↑ | PSNR↑ | SSIM↑ | PSNR↑ | SSIM↑ |
| UniPS Ikehata (2022) | 24.67 | - | - | - | - | - | - |
| IDarb Li et al. (2025) | 29.51 | 25.11 | 0.9271 | 25.62 | 0.9148 | 24.85 | 0.9342 |
| SDM UniPS Ikehata (2023) | 10.25 | 24.04 | 0.9076 | 23.68 | 0.9060 | 23.87 | 0.9289 |
| Uni MS-PS Hardy et al. (2024) | 8.92 | - | - | - | - | - | - |
| Ours | **4.51** | **30.49** | **0.9529** | **32.99** | **0.9357** | **31.32** | **0.9670** |

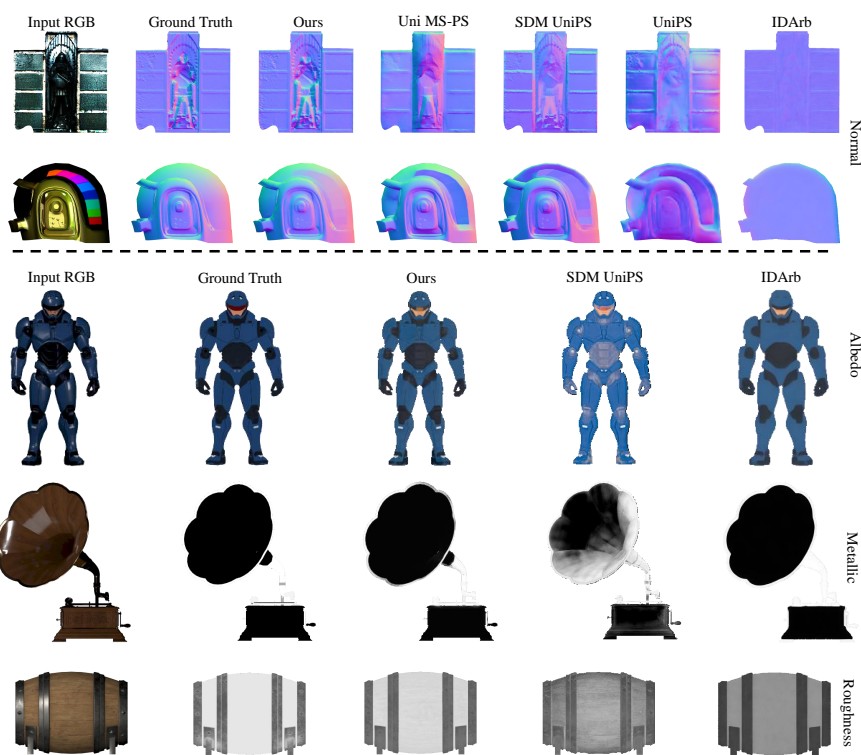

Figure A2: Qualitative comparison on PS-Verse Testdata. LINO UniPS demonstrates superior PBR estimation compared to all other methods.

representations: $\mathbf{x}_{\mathrm{env}}^{h}$, $\mathbf{x}_{\mathrm{point}}^{h}$, and $\mathbf{x}_{\mathrm{direction}}^{h}$, all in $\mathbb{R}^{D}$. Specifically, this projection is realized using three structurally similar two-layer Multi-Layer Perceptrons (MLPs). Within this common embedding space, we employ cosine similarity to supervise and align their respective feature distributions. Cosine similarity is chosen as it effectively measures the directional concordance between feature vectors, making it suitable for aligning representations of different lighting characteristics. This supervision translates into three distinct loss functions, denoted as $\mathcal{L}_{\mathrm{env}}$, $\mathcal{L}_{\mathrm{point}}$, and $\mathcal{L}_{\mathrm{direction}}$. Their respective mathematical formulations are:

$$\mathcal{L}_{\mathrm{env}} = 1 - \sum (\mathbf{l}_{\mathrm{env}}^{h} \cdot \mathbf{x}_{\mathrm{env}}^{h}) \tag{A1a}$$

$$\mathcal{L}_{\mathrm{point}} = 1 - \sum (\mathbf{l}_{\mathrm{point}}^{h} \cdot \mathbf{x}_{\mathrm{point}}^{h}) \tag{A1b}$$

$$\mathcal{L}_{\mathrm{direction}} = 1 - \sum (\mathbf{l}_{\mathrm{direction}}^{h} \cdot \mathbf{x}_{\mathrm{direction}}^{h}) \tag{A1c}$$

### A1.1.4 FUSION BLOCK:

The subsequent discussion details the Fusion Block applied to features extracted during the initial stages of our Encoder. The overall feature fusion process is DPT-based Ranftl et al. (2021). For clarity,

we will first illustrate this process using the features derived from the downsample image features $F_{d,f,r}^d$ as the primary example. Within each Interleaved Attention Block of the encoder, features obtained from its four internal attention—Frame, Light Axis, Global, and Light Axis—are first concatenated along the feature dimension. This operation yields an aggregated feature set for each block, denoted as $F_{d,f}^{d,(i)} \in \mathbb{R}^{L \times 4D}$, where $i \in \{1, 2, 3, 4\}$ represents the index of the $i$-th attention block. It is crucial to note that the three additional Light Register Tokens (introduced previously) do not participate in this specific feature aggregation (concatenation) process. Therefore, the feature dimension of $F_{d,f}^{d,(i)}$ is $4D$, not $4D'$.

Then, to effectively fuse features from different depths within the encoder, our Fusion Block employs a top-down, multi-scale fusion strategy. This process begins by selecting the aggregated output features, $F_{d,f}^{d,(i)}$, from four different stages of the Interleaved Attention Blocks. These features are initially 1D token sequences. First, these token sequences are reshaped from their 1D sequence format back into 2D spatial feature maps. Next, these four distinct spatial maps (which come from different depths of the Interleaved Attention Blocks) undergo a series of projection and downsampling operations to transform them into a standardized, four-level hierarchical feature pyramid, denoted $H_1, H_2, H_3$, and $H_4$. These operations typically consist of 1x1 convolutions to project the features to their target channel dimensions ($C, 2C, 4C, 4C$) and downsampling (2x2 convolutions with a stride of 2) to match the target resolutions. This pyramid captures multi-scale information, ranging from high-resolution features at $H_1$ (shape $F \times C \times H/2 \times W/2$) to low-resolution/high-semantic features at $H_4$ (shape $F \times 4C \times H/16 \times W/16$), where $F$ is the number of multi-light images, $C$ is 256, and $H, W$ refer to the spatial resolution of the original, full-sized input images. We then employ a progressive, top-down fusion path. This fusion is implemented using residual convolutional blocks He et al. (2015) and upsampling operations (2x2 transposed convolution with a stride of 2). Specifically, the deepest feature $H_4$ is first upsampled and then fused with $H_3$ via a residual block; this result is then upsampled and fused with $H_2$, and so on. The final output of this progressive fusion is a single, information-rich fused feature map, $F_{\text{fused}}^d$, with a shape of $\mathbb{R}^{F \times \frac{H}{2} \times \frac{W}{2} \times C}$, which is then passed to the Wavelet UpSample module. A similar fusion process is applied to the features derived from the wavelet components path ($F_{d,f,r}^w$), yielding a corresponding fused representation, $F_{\text{fused}}^w \in \mathbb{R}^{4 \times F \times \frac{H}{2} \times \frac{W}{2} \times C}$. It is noteworthy that this strategy of selecting four feature levels for hierarchical fusion can be adapted if more than four Interleaved Attention Blocks are employed in the encoder's initial stages. For instance, if six such blocks are utilized, features from blocks indexed 1, 2, 4, and 6 might be selected to form the pyramid. Similarly, for an eight-block configuration, features from blocks 1, 3, 5, and 7 could be chosen as inputs to the hierarchical fusion pathway.

### A1.1.5 Wavelet UpSample Module:

To obtain the final encoder output $F_{\text{enc}} \in \mathbb{R}^{F \times H \times W \times C}$, the features derived from the downsample image path $F_{\text{fused}}^d$ and those from the wavelet components path $F_{\text{fused}}^w$ should be integrated. The process is as follows: first, the feature map $F_{\text{fused}}^d$ is upsampled, yielding a representation $F_{\text{fused}}^{\text{up}} \in \mathbb{R}^{F \times H \times W \times C}$. Concurrently, for $F_{\text{fused}}^w$, which originates from the wavelet-transformed inputs, an inverse wavelet transform is applied to convert it back to the spatial domain, resulting in $F_{\text{fused}}^{\text{dwt}} \in \mathbb{R}^{F \times H \times W \times C}$. Finally, these two processed feature sets, $F_{\text{fused}}^{up}$ and $F_{\text{fused}}^{dwt}$, are element-wise summed. A Gaussian blur is subsequently applied to this sum to promote a smoother and more effective fusion of these potentially cross-domain features, ultimately producing the final encoder representation $F_{\text{enc}} \in \mathbb{R}^{F \times H \times W \times C}$.

### A1.1.6 Decoder:

The decoder architecture in our LINO UniPS is largely identical to that of SDM UniPS Ikehata (2023); for clarity, we briefly outline its key components and rationale here.

A common initial step in PS for surface normal estimation is the pixel-wise aggregation of spatial-light features along the illumination axis, effectively reducing $F$ light channels to a single representation per pixel using input images $I_f$ and their corresponding encoded features $F_{\text{enc}}^f$. We introduce an approach, termed the pixel-sampling Transformer Wu et al. (2024b); Contributors (2023); Wu et al. (2022; 2024a), which uniquely operates on a fixed count ($m$, e.g., $m = 2048$) of randomly chosen pixel locations. This strategy offers distinct advantages: it maintains a constant memory footprint

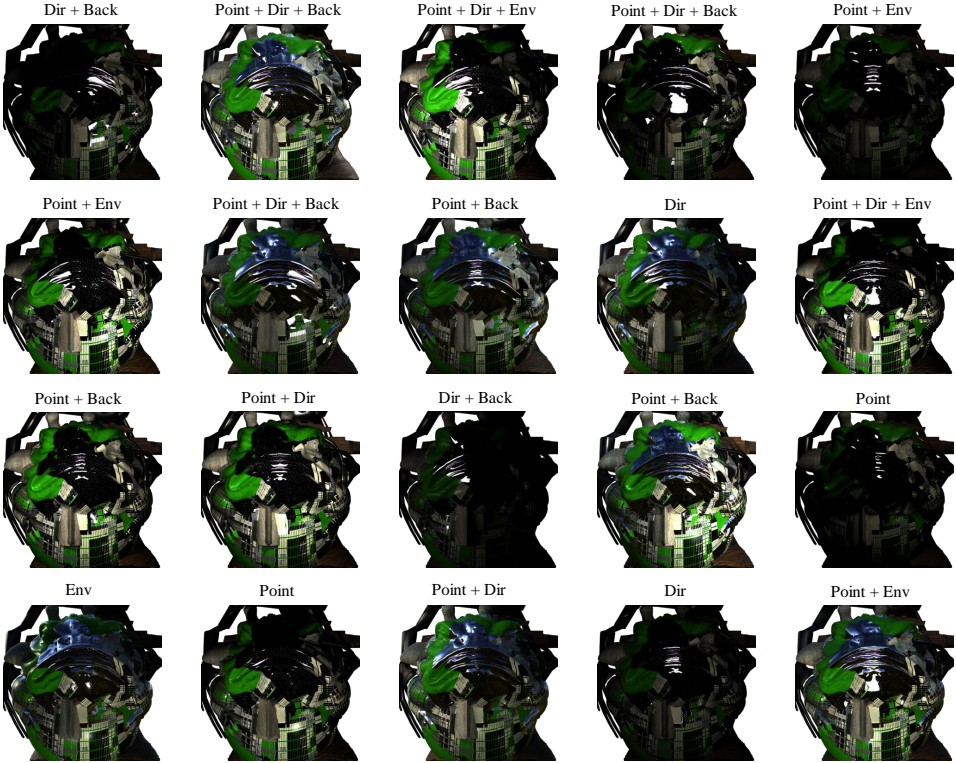

Figure A3: 20 multi-light images of one scene and their respective lighting configurations.

per sample set regardless of image dimensions, thus ensuring excellent scalability; furthermore, by processing a sparse, randomly distributed set of points, it substantially curtails over-smoothing artifacts often prevalent in dense convolutional operations. The practical implementation of the *pixel-sampling Transformer* involves selecting $m$ random pixels, denoted $\{x_i\}_{i=1}^m$, from the valid (masked) region of the input image. For each sampled pixel $x_i$, its associated features $F_{\text{enc}}^f(x_i) \in \mathbb{R}^C$ are obtained. These features $F_{\text{enc}}^f(x_i)$ are then combined through element-wise addition with $I_f'(x_i) \in \mathbb{R}^C$. The term $I_f'(x_i)$ represents a high-dimensional projection of the raw pixel observations $I_f(x_i) \in \mathbb{R}^3$, and this projection is performed by a two-layer MLP with the objective of enhancing the representational power of these raw observations by mapping them to this higher-dimensional space. Notably, our strategy of first projecting the raw observations to $\mathbb{R}^C$ and then performing addition differs from SDM UniPS Ikehata (2023), which typically employs direct concatenation of features and the raw observations. These added per-pixel features $F_{\text{add}}^f(x_i)$ are subsequently condensed into compact descriptors $A(x_i)$ by employing Pooling by Multi-head Attention (PMA) Lee et al. (2019). The resulting collection of $m$ descriptors, $A(x_i)_{i=1}^m$, is then fed into a Transformer network. The processing for each of the $m$ sampled points involves applying frame attention and light-axis attention to aggregate non-local context and cross-image information. Following this, a two-layer MLP is utilized to predict the surface normal vector for each location. These sparsely predicted normals are then systematically merged—for instance, through spatial interpolation or a dedicated upsampling module—to reconstruct the full-resolution surface normal map corresponding to the original input image dimensions. In essence, the pixel-sampling Transformer facilitates the modeling of robust non-local dependencies with notable computational efficiency, while concurrently preserving fine details in the output normal map. This makes the approach particularly well-suited for physics-based vision tasks Chen et al. (2025) that involve high-resolution imagery.

### A1.1.7 TRAINING LOSS:

In this part, we elaborate on the composition of our total training loss and the design of the respective weights for its constituent components. Based on the definitions provided in Eq. 3, Eq. 2 and Eq. A1, the overall training loss $\mathcal{L}$ for our LINO UniPS method can be expressed as:

$$\mathcal{L} = \lambda_1 \mathcal{L}_{\text{env}} + \lambda_2 \mathcal{L}_{\text{point}} + \lambda_3 \mathcal{L}_{\text{direction}} + \lambda_4 \sum (N - \tilde{N})^2 \odot C + \lambda_5 \sum (\tilde{G} - G)^2, \qquad \text{(A2)}$$

Let us define the confidence-weighted normal reconstruction loss as $\mathcal{L}_{\text{conf}}$ and the normal gradient supervision loss as $\mathcal{L}_{\text{g}}$. The overall training loss $\mathcal{L}$ can then be expressed as:

$$\mathcal{L} = \lambda_1 \mathcal{L}_{\text{env}} + \lambda_2 \mathcal{L}_{\text{point}} + \lambda_3 \mathcal{L}_{\text{direction}} + \lambda_4 \mathcal{L}_{\text{conf}} + \lambda_5 \mathcal{L}_{\text{g}} \qquad \text{(A3)}$$

Since our primary objective is surface normal reconstruction, $\mathcal{L}_{\text{conf}}$ (our confidence-weighted reconstruction loss) is established as the principal component of our total loss function. First, we provide a detailed explanation for our selection of $\mathcal{L}_{\text{conf}}$ as the primary loss function. We elaborate on why this specific formulation was chosen over other potential candidates, such as a direct MSE, $\sum (N - \tilde{N})^2$, or an alternative loss weighted by $e^G, G = \nabla N$, namely $\sum (N - \tilde{N})^2 \odot e^G$.

A primary motivation for our LINO UniPS framework is to advance beyond prior Universal PS methods by specifically improving the handling of challenging high-frequency regions. We identify these regions based on large magnitudes of the surface normal gradients, as these directly reflect geometric complexity. We deliberately avoid using gradients derived from the input RGB multi-light images as the primary criterion for this identification. The rationale is that while RGB gradients are indeed large in areas of intricate geometric detail, they can also exhibit high magnitudes in regions with significant basecolor variations, which do not necessarily correspond to the geometric high-frequency features we aim to emphasize and reconstruct accurately. Consequently, our methodology incorporates a loss function that is directly informed by surface normal gradients, rather than relying on a naive MSE.

A crucial aspect of this gradient-informed loss strategy concerns the source of the gradients utilized for weighting or guidance. We opt to utilize network-estimated normal gradients $\tilde{G}$ for this purpose, rather than directly employing ground truth normal gradients $G$. This design choice is primarily motivated by two factors: Firstly, it compels the network to intrinsically estimate high-frequency components from the input, thereby fostering its inherent capability to process and represent fine-grained details. Secondly, refraining from direct weighting by ground truth normal gradients typically leads to a more stable and manageable training process, especially during the initial stages when network predictions may significantly deviate from the ground truth.

Our design for the loss weights is as follows:

$$\lambda_1 = \frac{0.1}{(\mathcal{L}_{\text{env}}/\mathcal{L}_{\text{conf}})_{\text{sg}}}, \lambda_2 = \frac{0.1}{(\mathcal{L}_{\text{point}}/\mathcal{L}_{\text{conf}})_{\text{sg}}},$$
$$\lambda_3 = \frac{0.1}{(\mathcal{L}_{\text{direction}}/\mathcal{L}_{\text{conf}})_{\text{sg}}}, \lambda_4 = 1, \lambda_5 = \frac{0.1}{(\mathcal{L}_{\text{g}}/\mathcal{L}_{\text{conf}})_{\text{sg}}}$$

(A4)

where the subscript 'sg' denotes that the term within the parenthesis is treated as a constant (i.e., its gradient is not computed during backpropagation for the purpose of this scaling factor, akin to `.detach()` in PyTorch).

Our decision to set $\lambda_4$ to 1 is because $\mathcal{L}_{\text{conf}}$ serves as the principal component in our overall loss function. The remaining auxiliary losses are then scaled using the adaptive weighting mechanism detailed in Eq. A4. This mechanism constrains their magnitudes to 0.1 times that of the primary loss's detached value, $(\mathcal{L}_{\text{conf}})_{\text{sg}}$, while still allowing their gradients to backpropagate fully. Such a strategy effectively positions these auxiliary losses to act as regularizers to the main learning objective, rather than allowing disparate loss magnitudes to vie for dominance and potentially destabilize training. This controlled weighting is crucial for ensuring stable and efficient training of LINO UniPS, mitigating issues such as excessively slow convergence or even training failure that can arise from an unbalanced multi-term loss function.

## A1.2 PBR MATERIALS PREDICTION

The intrinsic properties of a surface (albedo, metallic, normal, and roughness) are fundamentally entangled, as they are all governed by the underlying surface geometry and material composition. Consequently, our LINO UniPS architecture, designed for normal recovery, can be naturally extended to jointly estimate a full set of PBR material parameters. This extension requires only finetuning on an appropriate dataset with a modified loss function.

### A1.2.1 IMPLEMENTATION DETAILS

Our PS-Verse dataset was created with this purpose in mind; alongside surface normals, we also rendered ground truth maps for albedo (3-channel RGB), metallic (1-channel), and roughness (1-channel). A sample of this data is shown in Sec. A1.3.

Starting with our LINO UniPS model pre-trained for normal estimation, we finetune it on this complete PBR dataset for approximately 10 epochs until convergence. This process enables the model to simultaneously predict all four properties from the same input images. For the loss function, we augment the original loss $\mathcal{L}$ with a weighted MSE term for the material maps. The new loss is defined as: $\mathcal{L}_{\text{PBR}} = \mathcal{L} + \lambda_6 \mathcal{L}_a + \lambda_7 \mathcal{L}_m + \lambda_8 \mathcal{L}_r$, where $\mathcal{L}_a, \mathcal{L}_m, \mathcal{L}_r$ are the MSE between ground truth and predicted albedo, metallic, and roughness maps, respectively. The coefficients $\lambda_6, \lambda_7, \lambda_8$ are their corresponding loss weights, which are defined as:

$$\lambda_6 = \frac{1}{(\mathcal{L}_a/\mathcal{L}_{\text{conf}})_{\text{sg}}}, \lambda_7 = \frac{1}{(\mathcal{L}_m/\mathcal{L}_{\text{conf}})_{\text{sg}}}, \lambda_8 = \frac{1}{(\mathcal{L}_r/\mathcal{L}_{\text{conf}})_{\text{sg}}} \tag{A5}$$

### A1.2.2 EXPERIMENTAL SETUP

**Baselines:** Besides the Universal PS methods mentioned in the main paper (UniPS Ikehata (2022), SDM UniPS Ikehata (2023), Uni MS-PS Hardy et al. (2024)), we added one more baseline, IDArb Li et al. (2025). It should be noted that we chose the BRDF version of SDM UniPS, which can predict PBR materials. And IDArb can also predict PBR materials. In contrast, UniPS and Uni MS-PS can only predict normals.

**Evaluation Metrics:** We evaluate our results using three metrics. For surface normal accuracy, we measure the Mean Angular Error (MAE, in degrees, ↓), consistent with our main experiments. For the predicted albedo, metallic, and roughness maps, we assess their quality via the Peak Signal-to-Noise Ratio (PSNR, in dB, ↑) and the Structural Similarity Index (SSIM ↑).

**Evaluation Dataset:** The evaluation is conducted on our PS-Verse Testdata, which contains ground truth for all four intrinsic properties.

### A1.2.3 EXPERIMENT RESULTS

The quantitative results for PBR material estimation are presented in Tab. A1. The results clearly show that our PBR-version LINO UniPS achieves SOTA performance across all predicted maps, including normal, albedo, metallic, and roughness. This comprehensive superiority further validates the effectiveness of our proposed architecture. Furthermore, we observe interesting trade-offs among the baseline methods. While IDArb underperforms all other baselines on the normal prediction task, it surpasses SDM UniPS in estimating the other PBR properties (albedo, metallic, and roughness).

Fig. A2 presents the qualitative results on our PS-Verse Testdata, where our LINO UniPS predictions demonstrate the highest visual fidelity to the ground truth for all properties. In contrast, the baselines exhibit noticeable artifacts. For complex normal estimation, most methods struggle, and IDArb fails completely by predicting a flat surface. For the other material maps, SDM UniPS tends to bake in RGB texture details (e.g., the barrel pattern), while IDArb fails to disentangle complex shading, particularly in shadowed regions like on the soundbox. These visual comparisons further highlight the robustness and superior disentanglement capabilities of our approach.

### A1.3 DATASET ANALYSIS AND PRESENTATION

### A1.3.1 CATEGORIZATION METHODOLOGY

To rigorously evaluate and enhance the capability of our LINO UniPS method for reconstructing surface normals of objects that feature high-frequency geometric details on complex surfaces, we curated a dedicated set of objects exhibiting diverse geometric complexities. These objects were subsequently graded by difficulty into five distinct levels, designated Level 1 to Level 5. Specifically, Levels 1–4 are classified following Dora Chen et al. (2024) criterion, based on the number of salient edges $N_\Gamma$. Level 5, in contrast, is distinguished by the use of normal mapping in its rendering. The specific criteria for this classification are as follows:

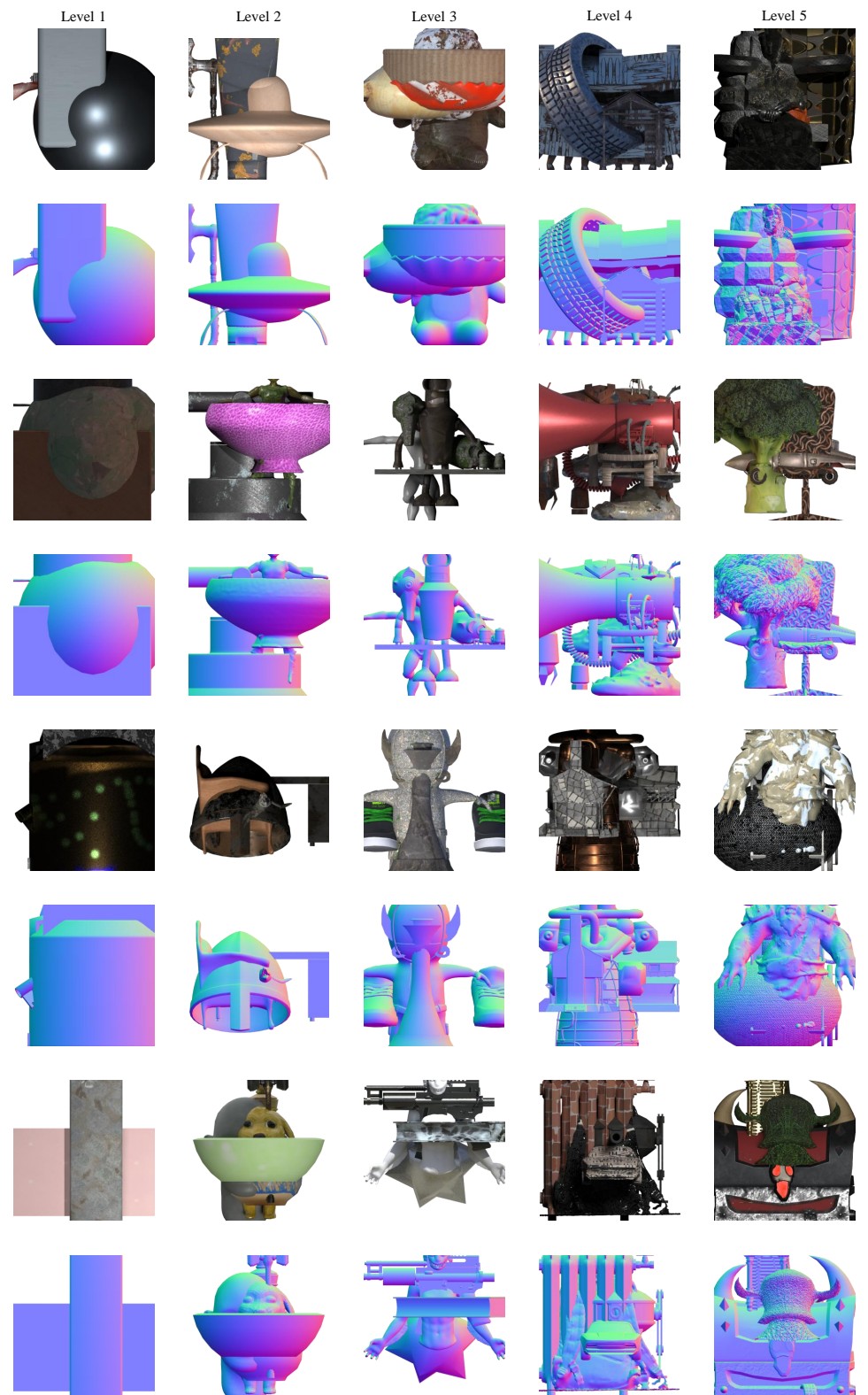

Figure A4: The objects are displayed sequentially from left to right, representing Level 1 to Level 5. Across Levels 1 through 5, there is a progressive rise in geometric complexity. Specifically, Level 5 features exceptionally complex surface geometry due to the utilization of normal mapping in its rendering process.

- Level 1 (Less Detail): $0 < N_\Gamma \le 5000$;

- Level 2 (Moderate Detail): $5000 < N_\Gamma \le 20000$;

- Level 3 (Rich Detail): $20000 < N_\Gamma \le 50000$;

- Level 4 (Very Rich Detail): $N_\Gamma > 50000$.

- Level 5 : With Normal Mapping

Fig. A4 shows representative cases from the different defined levels. PS-Verse comprises 100,000 scenes. For each of these scenes, two distinct renderings are typically generated: one that utilizes normal mapping to incorporate fine geometric details, and another rendered without this normal mapping. Levels 1-4 consist exclusively of scenes rendered without normal mapping, with each of these four levels containing 25,000 scenes. Level 5 is composed entirely of the 100,000 scenes rendered with normal mapping enhancement.

### A1.3.2  THE USE OF NORMAL MAPPING

To enhance PS normal reconstruction for objects characterized by intricate, high-frequency details, training data rich in such geometric features is essential. However, 3D models genuinely possessing fine-grained geometric intricacies are often scarce and prohibitively expensive, which impedes the creation of diverse, large-scale, high-fidelity datasets. To overcome this limitation within the Universal PS framework, our work pioneers the integration of normal mapping directly into the dataset generation process. Normal mapping, a 3D computer graphics technique, imbues low-polygon models with the visual appearance of high-frequency geometric details by applying a specialized texture, which encodes fine-scale perturbations of the surface normals. During rendering, these stored normal variations are then utilized to simulate intricate surface details without actually increasing the underlying geometric complexity or polygon count of the model.

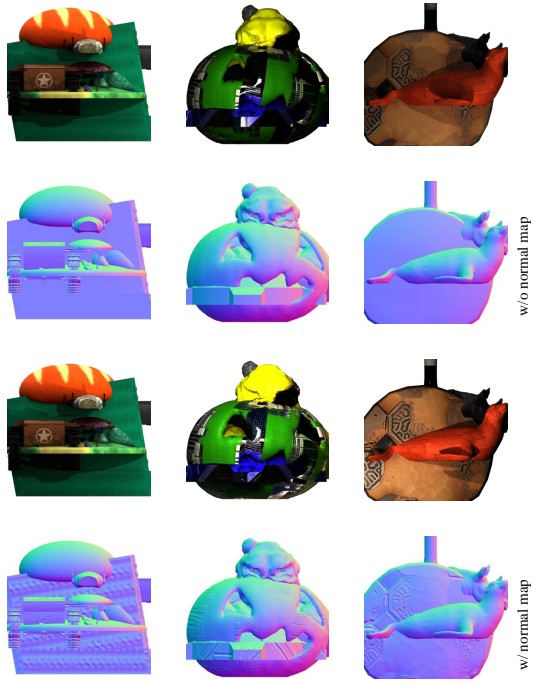

Figure A5: Effect of normal mapping on rendered surface detail. The top two rows display renderings without normal mapping, while the bottom two rows showcase the same scenes rendered with normal mapping. It is evident that employing normal mapping (bottom rows) results in significantly more high-frequency surface normal detail compared to renderings without (top rows).

A visual comparison of renderings with and without the use of normal mapping is presented in Fig. A5. It is clearly evident from the figure that employing normal mapping during the rendering process yields a significantly higher level of detail in the resulting surface normals.

### A1.3.3  LIGHTING SETUP

When rendering PS-Verse, we use four types of light sources; (a) environment lighting, (b) directional lighting, (c) point lighting, (d) uniform background lighting.

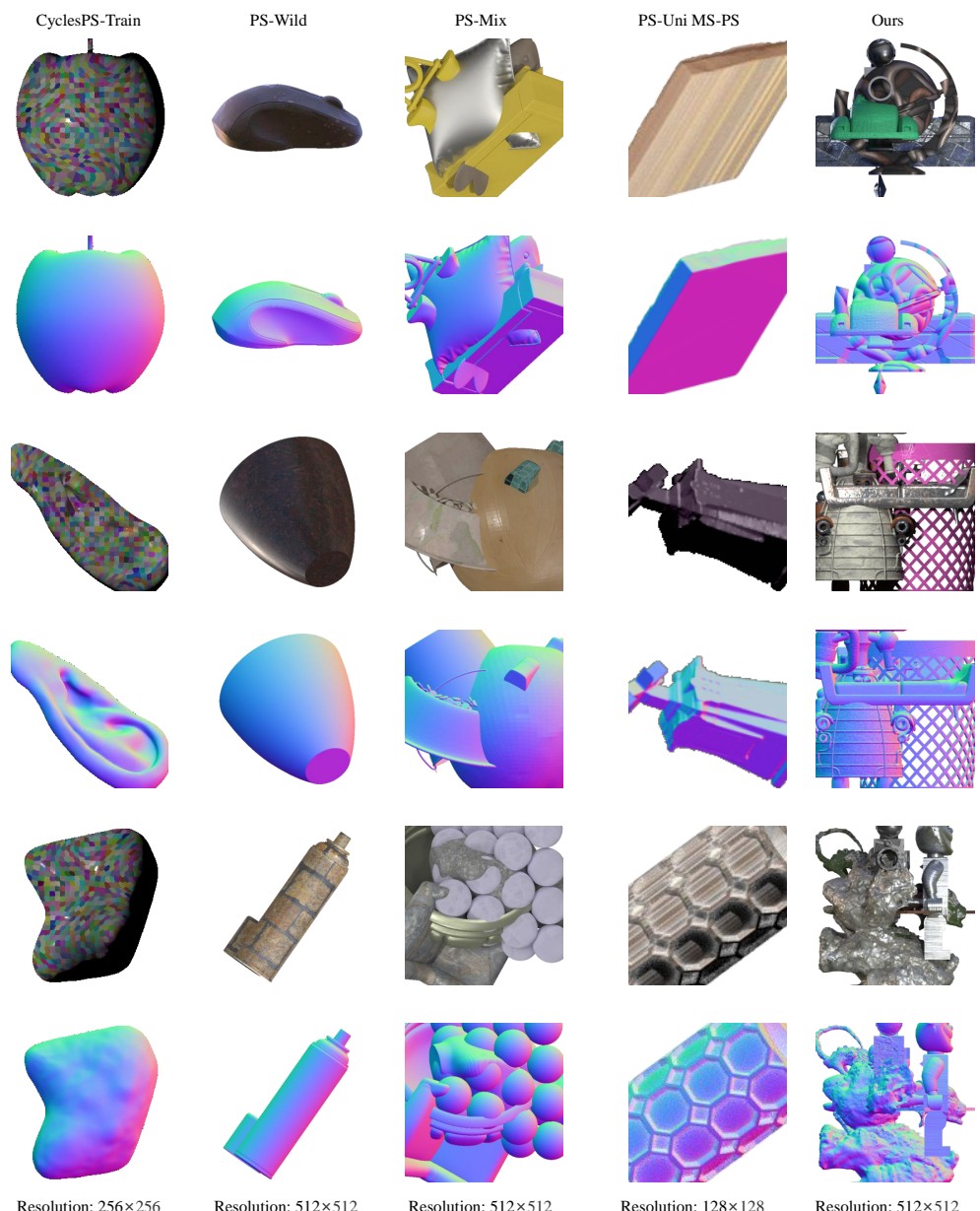

Figure A6: Visual comparison of different datasets. The spatial resolution of the images corresponding to each column is indicated beneath it.

During rendering, we generate ten distinct lighting configurations by combining several base lighting components (conceptually denoted here as (a), (b), (c), and (d)). These specific configurations are as follows: (1) Component (a), (2) Component (b), (3) Component (c), (4) Components (a) + (b), (5) Components (a) + (c), (6) Components (b) + (c), (7) Components (a) + (b) + (c), (8) Components (a) + (d), (9) Components (b) + (d), (10) Components (a) + (b) + (d). The lighting setup includes: directional light, point light and uniform background lighting, which is introduced to better simulate realistic global illumination. Every scene in PS-Verse is rendered as 20 images, each employing a lighting setup randomly chosen from our ten predefined lighting configurations. An example of such an image set for a single scene is illustrated in Fig. A3.

### A1.3.4 COMPARISON WITH OTHER DATASETS

Here, we primarily compare several training datasets: CyclesPS-Train Ikehata (2018), PS-Wild Ikehata (2022), PS-Mix Ikehata (2023), PS-Uni MS-PS Hardy et al. (2024), and our PS-Verse. For

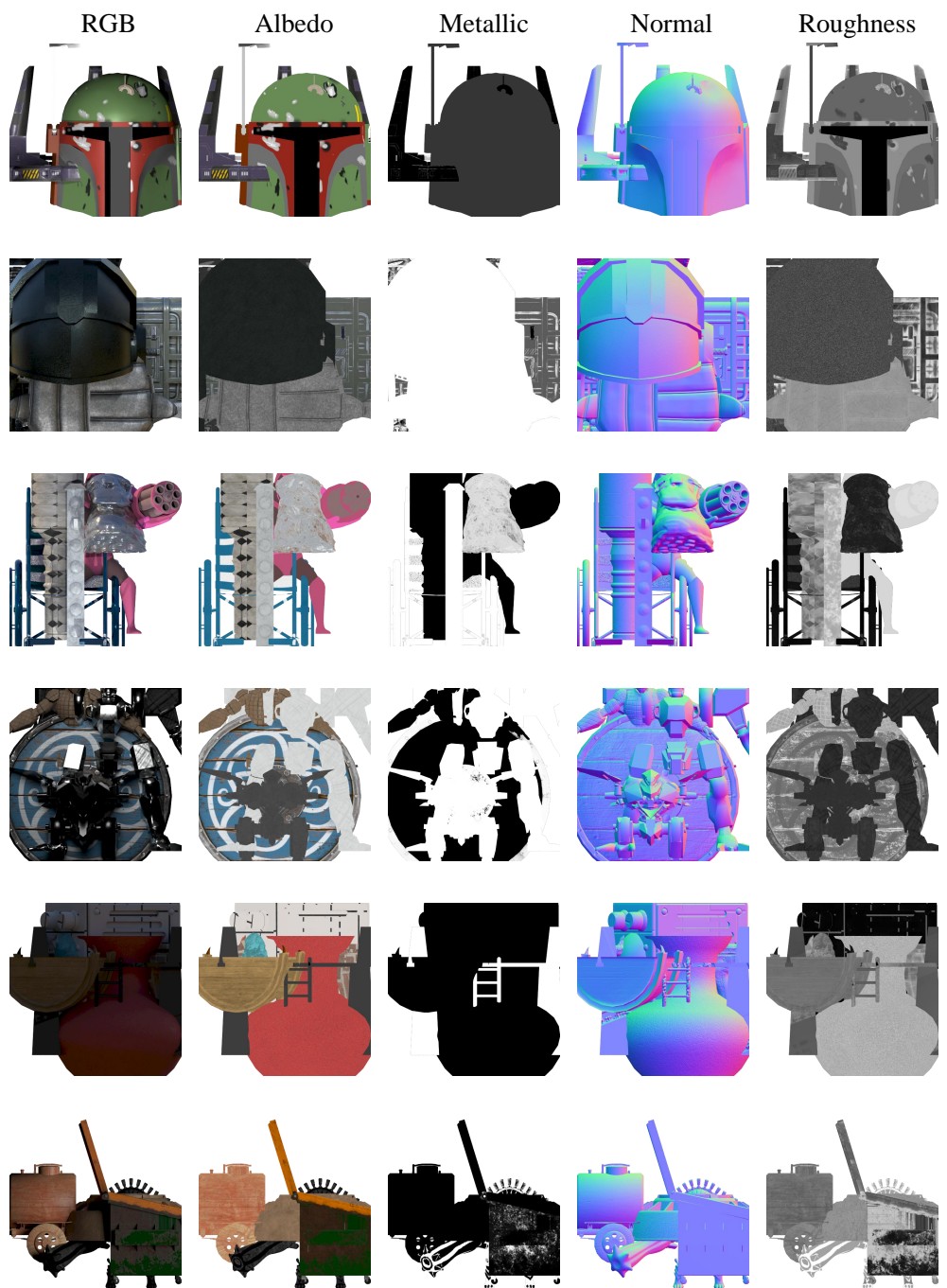

Figure A7: Our PS-Verse dataset contains ground truth for albedo, metallic, normal, and roughness.

a quantitative comparison of these datasets, please refer to Tab. 1. We now present illustrative qualitative comparisons in Fig. A6. Our comprehensive evaluation, encompassing both qualitative and quantitative aspects, leads us to conclude that PS-Verse is the premier training dataset in terms of quality for the Universal PS task. This emphasis on large-scale, high-quality data curation is also consistent with recent trends in dense correspondence and stereo matching Guo et al. (2025b;a).

### A1.3.5 PBR MATERIALS

In addition to ground truth normals, our PS-Verse dataset also provides ground truth maps for albedo, metallic, and roughness, as shown in Fig. A7.

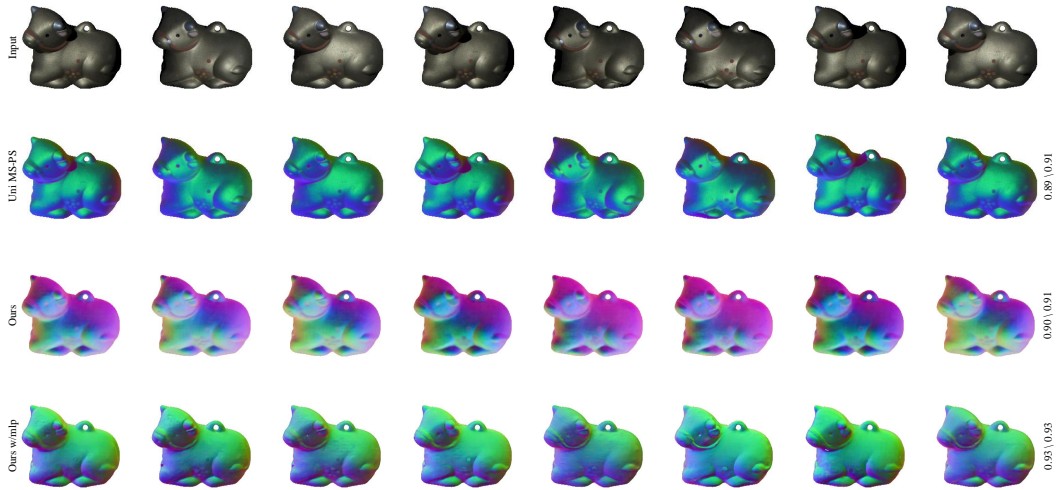

Figure A8: Post-PCA visualization of features extracted by different method encoders for the CowPNG from the DiLiGenT Shi et al. (2018). Metrics displayed to the right of each row are (CSIM/SSIM), higher values indicate higher feature similarity.

## A1.4 ADDITIONAL DISCUSSION

### A1.4.1 WHY FEATURE CONSISTENCY MATTERS

Universal PS aims to solve an inverse problem: recovering the intrinsic surface normal field $N$ from a set of observations $\{I_f\}_{f=1}^F$ formed under varying, unknown illumination conditions $\{\mathbf{L}_f\}_{f=1}^F$. The imaging process can be formally described as $I_f = \mathcal{F}(N, \rho, \mathbf{L}_f)$, where $\mathcal{F}$ represents the rendering equation and $\rho$ the surface reflectance.

An ideal encoder $E$ seeks to extract a unified feature representation $F_{enc} = E(\{I_f\}_{f=1}^F)$ that is strictly illumination-invariant, effectively marginalizing out the extrinsic variable $\{\mathbf{L}_f\}_{f=1}^F$. In this physical context, feature similarity (CSIM; SSIM) serves as a direct quantitative metric of this invariance; a low CSIM/SSIM score implies that the feature $F_{enc}$ retains significant dependency on $\{\mathbf{L}_f\}_{f=1}^F$, indicating a failure to decouple extrinsic illumination from intrinsic geometry.

The decoder $D$ is tasked with learning the mapping $\hat{N} = D(\{\mathbf{Z}_f\}_{f=1}^F)$. When features are not decoupled, the decoder confronts a highly ill-posed problem: it receives highly variable inputs $F_{enc}$ for the exact same physical geometry $N$, which inevitably introduces ambiguity and variance into the estimator, manifesting as higher reconstruction error (MAE).

Therefore, a primary driver for our LINO UniPS is the introduction of an improved encoder $E$. By explicitly employing Light Register Tokens supervised by Light Alignment to physically isolate variant illumination components $\{\mathbf{L}_f\}_{f=1}^F$, our encoder ensures that the feature representation passed to the decoder remains pure and consistent. This effectively mitigates the ill-posed nature of the decoding task, naturally yielding higher accuracy.

While the decoder's capabilities represent a non-negligible factor in overall performance, the decoders utilized in UniPS, SDM UniPS, and our LINO UniPS are architecturally similar, all adhering to the pixel-sampling paradigm. Consequently, the direct correlation between greater feature consistency and superior normal reconstruction is not an oversimplification; thus, when analyzing this relationship in Fig. 1, we group these three methods together to facilitate a direct comparison of the impact of their respective encoder-derived features.

Fig. A8 presents Principal Component Analysis (PCA) visualizations of features extracted by the encoders of various methods, alongside their corresponding feature similarity metrics (CSIM/SSIM). In the following discussion, we focus our analysis on our Ours w/mlp variant and Uni MS-PS.

Ours w/MLP refers to a configuration of our LINO UniPS where the standard decoder is replaced by a simple two-layer MLP. As illustrated in Fig. A8, this variant gives the highest feature similarity. Visual inspection of the PCA plots further reveals that its extracted features have effectively disentangled lighting information. We hypothesize that the superior feature similarity of Ours w/mlp compared to the full LINO UniPS (with its original decoder) stems from the constraints imposed by the weaker MLP decoder; this limited decoder capacity compels the encoder to learn more consistent features to facilitate accurate normal reconstruction. While this enhanced feature consistency from the encoder may not entirely compensate for the reduced representational power of the simpler decoder in terms of absolute normal reconstruction quality (when compared to the full LINO UniPS), Ours w/MLP variant nevertheless significantly outperforms SDM UniPS. This finding strongly corroborates our central hypothesis regarding the critical role of a powerful and well-regularized encoder in achieving effective feature disentanglement and consistency.

Uni MS-PS also demonstrates high feature similarity. However, visual analysis of its features (Fig. A8) suggests that they remain considerably entangled with lighting information. Consequently, we infer that its high reported feature similarity may be more attributable to geometric self-consistency within its representations rather than successful illumination decoupling. Despite this apparent lack of complete feature decoupling, Uni MS-PS often achieves commendable reconstruction results. We attribute this primarily to its multi-scale architecture: beyond the initial stage, each subsequent network stage in Uni MS-PS incorporates predicted normals from the preceding stage as an additional input, effectively leveraging them as a strong geometric prior. This iterative refinement, guided by intermediate normal predictions, places Uni MS-PS in a distinct operational paradigm compared to methods like UniPS, SDM UniPS, and our LINO UniPS.

Furthermore, we need to figure out that Uni MS-PS exhibits certain practical limitations. (a) Its multi-scale nature leads to considerable inference latency, particularly when processing multiple high-resolution input images (e.g., handling 16 images at 4K resolution can extend to about an hour). In contrast, our LINO UniPS method typically completes inference within tens of seconds for similar inputs (see Tab. 6). (b) While Uni MS-PS can reconstruct detailed surface normals, its reliance on potentially lower-resolution training datasets and its patch-based inference mechanism can lead to a loss of global contextual information, sometimes resulting in reconstructions that are locally detailed but globally inconsistent or erroneous (see Fig. 1 and Fig. 7) .

### A1.4.2    LIMITATIONS

Despite the commendable performance demonstrated by LINO UniPS, certain limitations remain, offering avenues for future research.

Firstly, the incorporation of global attention within our encoder, while designed to enhance inter-image feature interaction for more effective illumination-normal decoupling and successfully improving disentanglement, inevitably introduces additional computational burden. Consequently, a key direction for future work is to explore more computationally efficient mechanisms that can achieve comparable decoupling efficacy at a reduced operational cost.

Secondly, despite its strong generalization capabilities, it is important to acknowledge that the Universal PS paradigm exhibits certain inherent drawbacks compared to traditional paradigms.

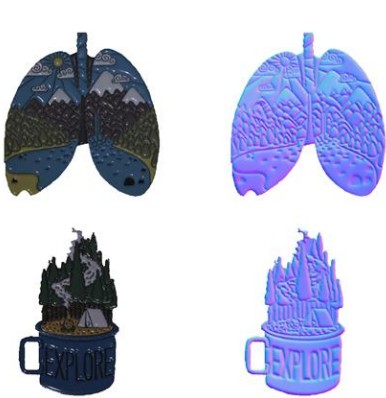

Figure A9: For near-planar objects possessing intricate concave and convex surface details, our LINO UniPS tends to invert the predicted surface normals. The objects are from DiLiGenT-II Wang et al. (2023)

For example, traditional Calibrated PS methods benefit from explicit, known light source parameters. In contrast, Universal PS lacks any explicit light source input (Although we employ Light Register Tokens to mitigate this, our network operates without explicit light source input during inference. However, Calibrated PS methods do utilize). Consequently, when handling near-planar objects, the Universal PS method struggles to unambiguously distinguish whether light originates from "above" or "below", resulting in the

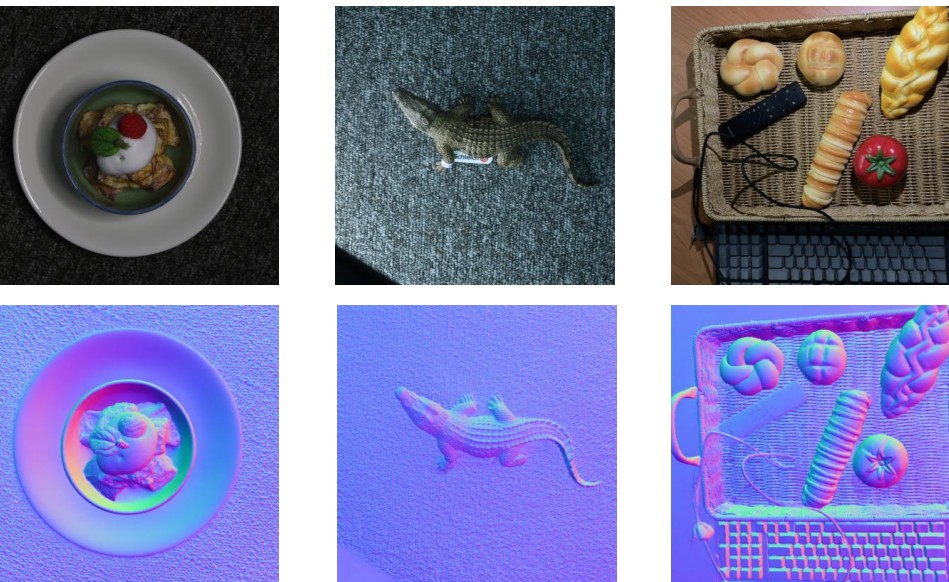

Figure A10: Top row: Example from the input multi-light images. Bottom row: Surface normal map reconstructed by our LINO UniPS.

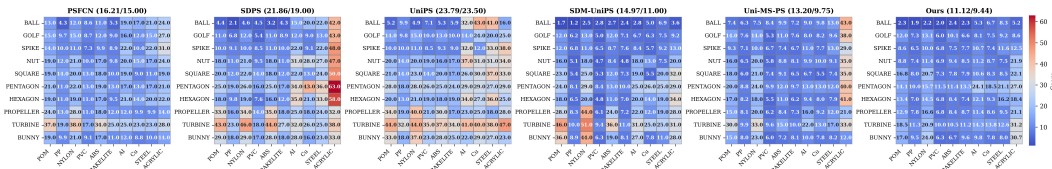

Figure A11: Results on the DiLiGenT10$^2$ dataset Ren et al. (2022): a matrix comparing performance where rows/columns corre- spond to shapes/materials. For enhanced detail visibility, viewing the electronic version in color is recommended.

reconstructed surface normals being inverted (see Fig. A9), whereas Calibrated PS handles this relatively well.

Furthermore, the Universal PS method is data-driven, inherently relying on massive amounts of data. Conversely, traditional Calibrated and Uncalibrated PS approaches do not rely as heavily on large-scale training datasets.

### A1.4.3   THE USE OF LARGE LANGUAGE MODELS (LLMS)

During the preparation of this manuscript, we utilized Large Language Models (LLMs), including Google's Gemini, as a writing assistant. The primary application of these models was for language enhancement tasks, such as improving grammar, refining phrasing for clarity, and ensuring stylistic consistency throughout the paper. It is important to note that the core scientific contributions, experimental design, results, and analyses presented herein were conceived and executed solely by the authors. The LLMs' role was strictly limited to that of a sophisticated tool for polishing the language and presentation of our work.

### A1.5   EXTENDED RESULTS

In this section, we present additional results to further demonstrate the capabilities of our LINO UniPS.

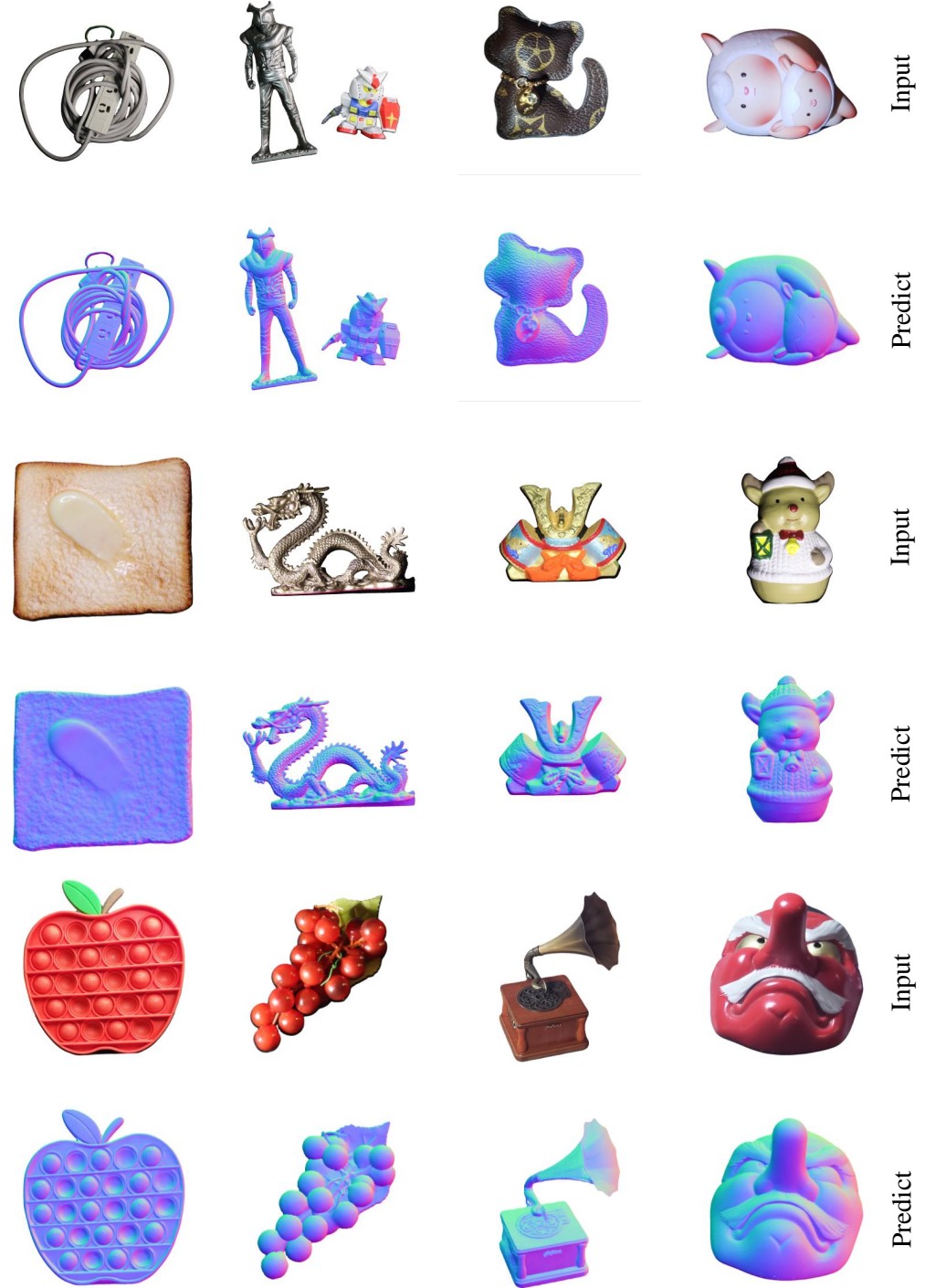

Figure A12: Real-world data with masks and corresponding LINO UniPS reconstruction results; data sourced from UniPS Ikehata (2022) and SDM UniPS Ikehata (2023).

### A1.5.1 PUBLIC BENCHMARKS

In this section, we present additional evaluation results on the DiLiGenT10$^2$ benchmark Ren et al. (2022), which are shown in Fig. A11. For a comprehensive comparison, we evaluate against a diverse set of five baselines spanning three categories: a representative Calibrated PS method (PS-FCN Chen et al. (2018)), a representative Uncalibrated PS approach (SDPS Chen et al. (2019)), and three

Universal PS methods (UniPS Ikehata (2022), SDM UniPS Ikehata (2023), and Uni MS-PS Hardy et al. (2024)). The results demonstrate that our LINO UniPS not only achieves the best performance among all Universal PS methods but also significantly surpasses the specialized Calibrated and Uncalibrated approaches. This superiority is particularly pronounced for objects with challenging material properties (e.g., ACRYLIC) or complex geometries (e.g., PENTAGON). In these demanding scenarios, our method's advantage over other contemporary Universal PS techniques becomes even more evident, highlighting its robustness.

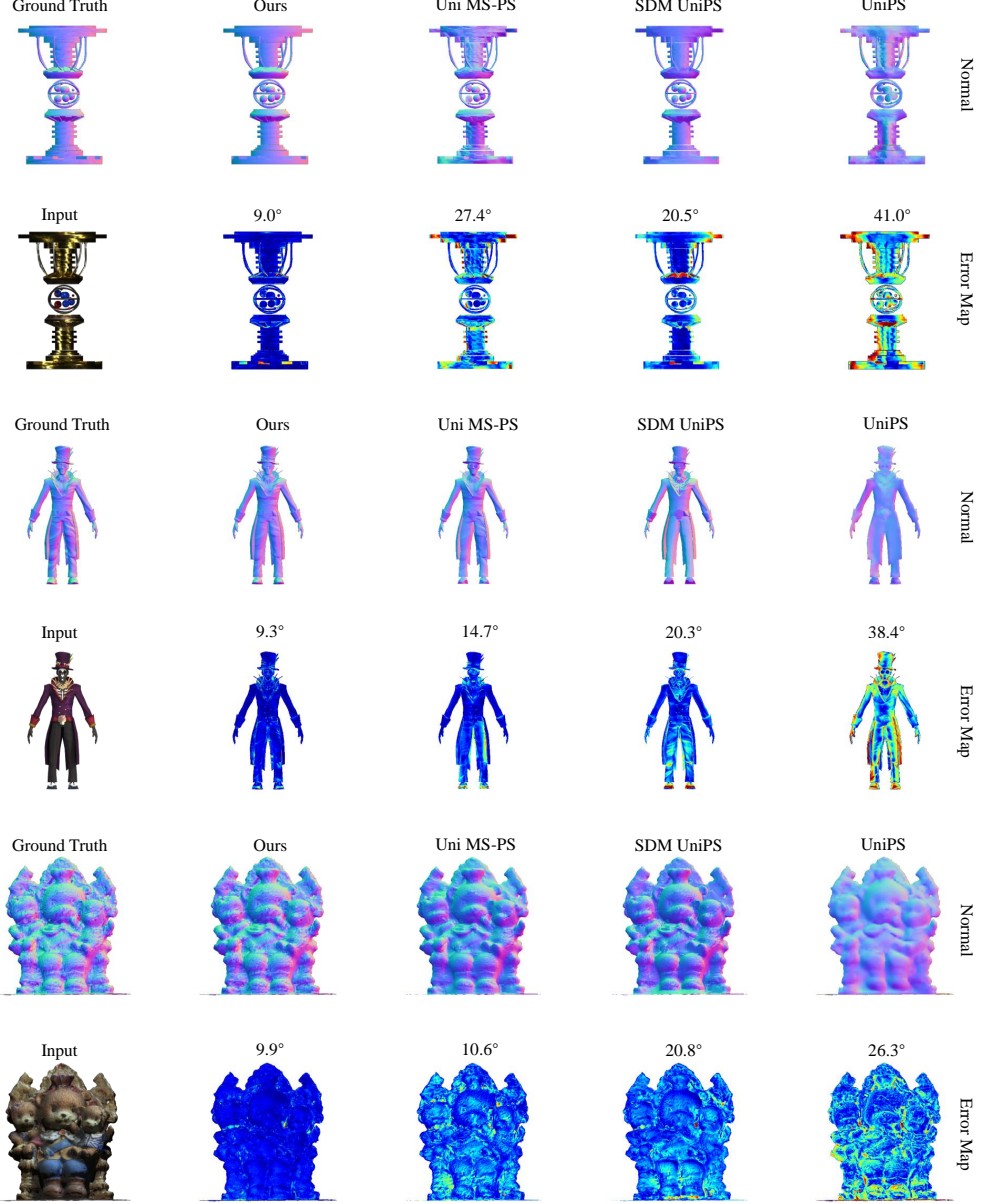

Figure A13: Comparison of different Universal PS methods on our PS-Verse Testdata, showcasing ground truth normals, reconstruction normals, and corresponding error maps. The error maps depict the Mean Angular Error (MAE), measured in degrees; lower MAE values signify a more accurate reconstruction.

### A1.5.2 REAL DATA & SYNTHETIC BENCHMARK

Fig. A10 showcases our method's performance on challenging real-world data. To demonstrate its robustness, the figure includes mask-free examples from two distinct sources: the two leftmost

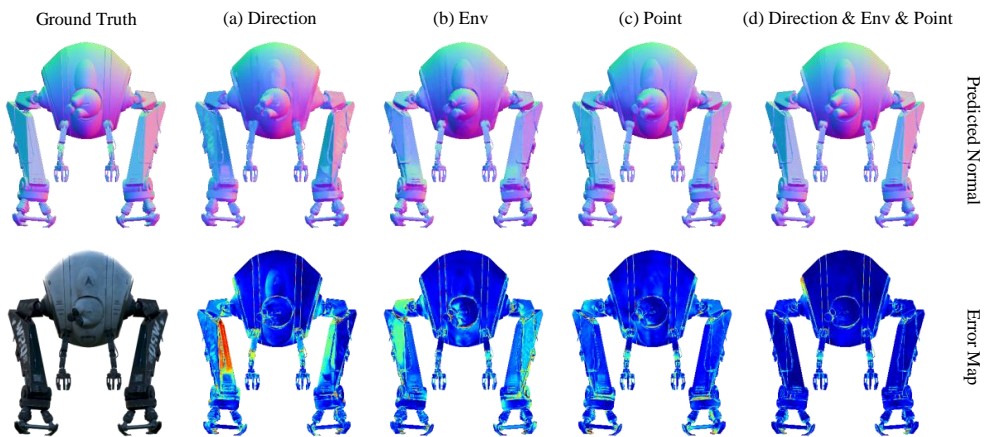

Figure A14: Qualitative results of the effect of each Light Register Token type. The full three-token model (right) achieves higher visual fidelity than any single-token variant, highlighting the need for all three.

columns show 4K resolution images from the SDM UniPS real data Ikehata (2023), while the rightmost column displays our own 960×960 captures using a mobile phone.

These results demonstrate that our LINO UniPS is robust to variations in scale and mask-free inputs, while consistently reconstructing fine-grained details.

Fig. A12 presents examples of real-world captured objects, the overall quality of these reconstructions underscores our approach's strong generalization capabilities.

Fig. A13 presents a comparison of various Universal PS methods on our PS-Verse Testdata. Given that PS-Verse Testdata is a synthetic dataset, ground truth is readily available, facilitating precise quantitative evaluation. The results clearly demonstrate that our LINO UniPS significantly outperforms contemporary approaches, including Uni MS-PS Hardy et al. (2024), SDM UniPS Ikehata (2023), and UniPS Ikehata (2022).

## A1.6 ADDITIONAL ABLATIONS

### A1.6.1 EFFECT OF EACH LIGHT REGISTER TOKEN TYPE

To analyze the independent contributions of each proposed Light Register Token, we designed a fine-grained ablation study. In this experiment, we use the final model presented in the last row of Tab. 2 as our baseline. The variable is the specific Light Register Token used, along with their corresponding light alignment losses. We evaluated the following four setups: (a) Using **Direction** Tokens, (b) Using **Env** Tokens, (c) Using **Point** Tokens, (d) Using **Point & Direction & Env** Tokens (final model).

Table A2: Quantitative comparison of single-token (Direction, Env, Point) performance against the full three-token combination, highlighting the Point token's contribution and the critical synergy of all three.

|  | CSIM↑ | SSIM↑ | Avg. MAE↓ |
|---|---|---|---|
| Direction | 0.84 | 0.81 | 5.32 |
| Env | 0.83 | 0.82 | 5.30 |
| Point | 0.86 | 0.84 | 4.98 |
| Direction & Env & Point | **0.88** | **0.86** | **4.51** |

The total number of register tokens was kept at three for all setups (consistent with the final model). All other experimental settings were identical.

Quantitative results are shown in Tab. A2, and qualitative results are in Fig. A14. We observe that: among all single-token configurations, using only the Point Tokens yields the most significant performance improvement, while the effects of the Direction and Env tokens are similar. This is because the Point Tokens represent high-frequency illumination information (such as specular highlights) , which provides the most critical and informative cues for resolving surface normals Ikehata (2023; 2022). Conversely, the Direction and Env tokens correspond to low-frequency light sources, providing relatively fewer geometric cues.

Ground Truth    Generic    Specialized

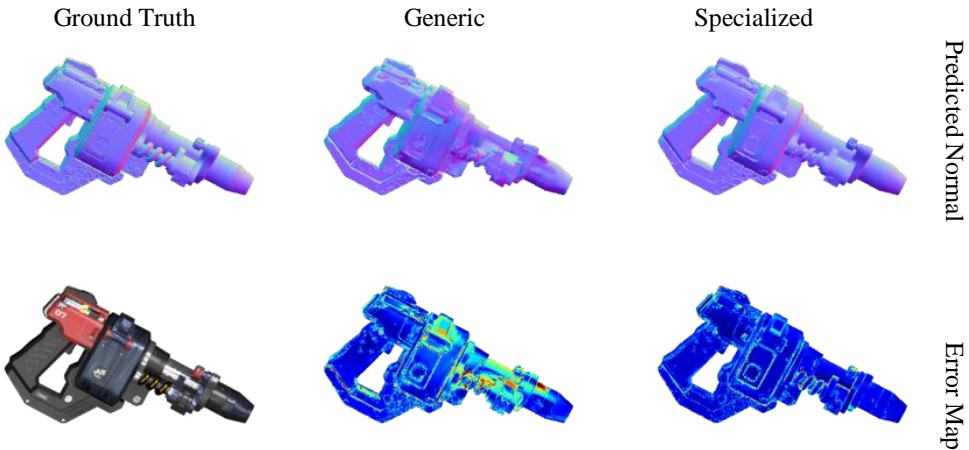

Figure A15: Qualitative comparison of Specialized vs. Generic Light Register Tokens. Our specialized token approach (right) exhibits superior visual fidelity.

Furthermore, we observe that a significant performance gap still exists between any single-token variant (configurations (a), (b), (c)) and the full model using all three tokens (configuration (d)). This strongly demonstrates that our design is not redundant: real-world illumination is a complex mixture, and the synergistic effect of all three tokens enables the model to handle more complex lighting conditions, thus most thoroughly decoupling the complex illumination information from the normal features.

### A1.6.2 WHY SPECIALIZED TOKENS INSTEAD OF GENERIC TOKENS

Table A3: Quantitative comparison of Specialized vs. Generic Light Register Tokens. Our specialized token design outperforms the generic token method.

|  | CSIM↑ | SSIM↑ | Avg. MAE↓ |
|---|---|---|---|
| Generic | 0.84 | 0.84 | 5.07 |
| Specialized | **0.88** | **0.86** | **4.51** |

We introduce three specialized Light Register Tokens to correspond to three distinct illumination types (Point, Direction, and Env). A natural question arises: why not use generic tokens instead of this specialized design? To answer this, we conduct an ablation study. For this experiment, we use our final model (last row of Tab. 2) as the baseline. All other experimental settings remain identical. The only change we make is to replace our three specialized Light Register Tokens with three generic tokens. These generic tokens are simultaneously supervised by all three light alignment losses ($\mathcal{L}_{env}$, $\mathcal{L}_{point}$, and $\mathcal{L}_{direction}$)

Quantitative results are shown in Tab. A3, and qualitative results are in Fig. A15. We find that the performance of the generic token approach is lower than our specialized token approach. We analyze that this is because the generic tokens are forced to entangle physically disparate lighting cues (e.g., high-frequency and low-frequency light), which consequently degrades the accuracy of the subsequent normal prediction.

### A1.6.3 WHY USING DUAL-BRANCH ARCHITECTURE

This section analyzes the advantages of our Dual-branch architecture (parallel wavelet and naive downsample branches). Quantitative results are in Tab. A4, and qualitative results are in Fig. A16. For this experiment, we use our final model (last row of Tab. 2) as the baseline. All other settings are identical; we only modify the encoder's feature extraction architecture to create three variants: (1) Downsample-only (2) Wavelet-only (3) Dual-branch (final model).

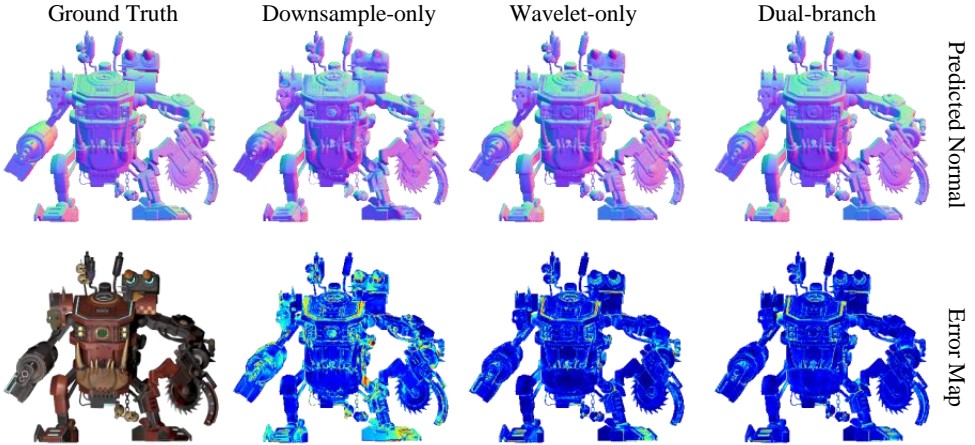

Figure A16: Qualitative comparison of Dual-branch model against its single-branch variants, confirming the complementary effect.

From both quantitative and qualitative results, we find that Wavelet-only outperforms Downsample-only. For normal reconstruction, naive downsample irreversibly discards critical high-frequency geometric information. In contrast, the wavelet transform, by explicitly preserving these high-frequency components , captures finer surface details and yields performance gain.

Furthermore, Dual-branch model outperforms all single-branch variants. Our analysis is as follows: The two branches achieve functional specialization. The wavelet branch focuses on explicitly preserving high-frequency, fine-grained local geometric details. The naive downsample branch, while lossy in detail, provides a smoother, more anti-aliased low-frequency representation, offering robust global image-domain semantic information Glasner et al. (2009); Zhang (2019); Liu et al. (2024); Yang et al. (2024). The advantages of both branches are complementary Guo et al. (2017); Zheng et al. (2023), and their fusion allows the model to achieve superior accuracy; therefore, despite the low-frequency component already included in the wavelet transform, we additionally retain the parallel naive downsample branch.

Table A4: Quantitative comparison of Dual-branch model against its single-branch variants, demonstrating the quantitative advantage of the Dual-branch design.

|  | CSIM↑ | SSIM↑ | Avg. MAE↓ |
|---|---|---|---|
| Downsample-only | 0.80 | 0.79 | 5.98 |
| Wavelet-only | 0.85 | 0.84 | 4.97 |
| Dual-branch | **0.88** | **0.86** | **4.51** |

