# OpenReview forum: "Light of Normals: Unified Feature Representation for Universal Photometric Stereo"
_ICLR.cc/2026/Conference — ICLR 2026 Poster_

### Official Review · Reviewer_nvYr · 2025-10-25

**Soundness:** 3
**Presentation:** 4
**Contribution:** 3
**Rating:** 8
**Confidence:** 3

**Summary:**

This paper proposes LINO UniPS, a transformer-based framework for Universal Photometric Stereo. It recovers surface normal maps from multiple images under unknown lighting conditions. They propose the Light Register Tokens and the Interleaved Attention Block to allow the model to understand normal and light conditions separately. Also, they propose the Wavelet-based Dual-branch Architecture and a normal-gradient perception loss to preserve high-frequency geometric details. They demonstrated their methods on PS-Verse, a new, high-quality, large-scale synthetic dataset they created.

**Strengths:**

- The explicit light-feature decoupling through LRTs and interleaved attention is well justified and ablated.
- Consistent performance improvements have been achieved, including in the benchmarks presented.
- The paper is well-organized.

**Weaknesses:**

- Some ablations (e.g., the effect of each light type token separately) could be presented in more detail.
- There are no qualitative results for ablation studies.

**Questions:**

- Looking at the Light Registered Attention module in Figure 2, it seems like frame tokens are copied and utilized for each of the three light registers. Am I understanding this correctly? This part seems like a huge computational burden. Why did you bother calculating the three separately?
- You claim that each of your methods solves the decomposing and high-frequency problems, but can you show qualitative ablation, not just performance improvement?

---

> ### Author Response · Authors · 2025-11-19
>
> We sincerely thank you for the time and effort dedicated to reviewing our paper. Your extremely professional and insightful review was critical in helping us improve the quality of our manuscript. We are extremely grateful for your high recognition of our work! The corresponding changes addressing your questions have been highlighted in **Purple** in the newly uploaded PDF.
>
> ---
> **W1: Some ablations (e.g., the effect of each light type token separately) could be presented in more detail.**
>
> This is a very valuable suggestion! We have conducted a detailed ablation study to analyze the specific effect of each light register token type. Please refer to **Sec. A.1.6.1** in our newly uploaded PDF.
>
> **W2: There are no qualitative results for ablation studies.**
>
> This is a critical point! We have added the qualitative results for the ablation study to visually demonstrate the efficacy of our method. Please refer to **Fig. 6** in the updated PDF.
>
> **Q1: Misunderstanding of the Light Registered Attention Module.**
>
> We sincerely apologize for the misunderstanding, which was likely caused by the lack of clarity in our previous Fig. 2. We would like to provide the following clarifications:
> * Each input image corresponds to its own specific set of frame tokens.
> * The three Light Register Tokens are concatenated with each image's frame tokens; the frame tokens themselves are **not copied**.
>  * Since there are very few Light Register Tokens (only three), this design introduces almost **no additional computational burden**.
>
> We have revised **Fig. 2** in the updated PDF to clearly illustrate this process and prevent further misunderstanding. We apologize again for the confusion.

---

### Official Review · Reviewer_zyYx · 2025-10-27

**Soundness:** 2
**Presentation:** 2
**Contribution:** 2
**Rating:** 2
**Confidence:** 3

**Summary:**

The paper proposes LINO UniPS, a method for universal photometric stereo. It first introduces light register tokens for different types of lightings and interleaved attention block to enable better separation of lighting and lighting-invariant normal in the encoder. Then it uses a Wavelet-based Dual-branch architecture and a normal-gradient perception loss to recover the the fine details of the scenes. Furthermore, it introduces a new dataset, PS-Verse, with more complicated surface and lightings for the photometric stereo task.

**Strengths:**

1. The proposed method achieves good results on two benchmark datasets, DiLiGenT and LUCES.

2. The paper introduces a new dataset, PS-Verse, which is proved to be able to help achieve better performance for the same method from Table 2.

**Weaknesses:**

1. The clarity of the paper could be further improved. In general I can understand the idea of the paper but there are several key aspects that I am confused with. Please see Questions section below. Also there are not metrics introduction for table 4 and table 5 in their table descriptions.

2. The test scenes only have one object per-scene. I wonder if the method can handle more complicated scenes?

3. My biggest concern is the comparison results with other methods.

(1) In Uni MS-PS (https://hal.science/hal-04431103v2/file/main_hal.pdf) Table 3, Uni MS-PS can achieve 6.04 when using 30 images for Buddha, which is better than LINO UniPS and not shown in Table 4. For POT1, Uni MS-PS can achieve 4.08 which is very similar to LINO UniPS. For COW, best Uni MS-PS is the same as LINO UniPS and also not shown in table 4.

(2) The same is for LUCES dataset, not the best results of Uni MS-PS are reported in table 5, e.g. for BOWL, BUDDHA.

**Questions:**

1. The training starts from PS-Verse Level 1 data to level 4 data, which don't have ground-truth normal maps, I wonder how do you compute $L_n$ for these data?

2. For the feature similarity metric (CISM), what are the two features to compare with?

3. In table 4, for POT2, SDM-UniPS seems to achieve better results?

---

> ### Author Response · Authors · 2025-11-19
>
> We sincerely thank the reviewer for the time and effort dedicated to reviewing our paper! Your comments are critical for improving the quality of our manuscript. We have highlighted the corresponding changes addressing your questions in **Green** in the updated PDF.
>
> ---
> **W1: The clarity of the paper could be further improved. Also, there are no metrics introduced for Table 4 and Table 5 in their table descriptions.**
>
> Thank you for pointing out this issue. We have revised the paper for better clarity. Additionally, we have added the metric (Mean Angular Error, MAE) description to the captions of **Tab. 4** and **Tab. 5**. Please refer to the updated PDF.
>
> **W2: I wonder if the method can handle more complicated scenes?**
>
> Absolutely! Please refer to the results in **Fig. 7, Fig. A10, and Fig. A12** in the updated PDF. These figures demonstrate the method's capability to handle complex cases, including scenes with multiple objects.
>
> **W3: Comparison results with other methods (Uni MS-PS).**
>
> The captions for Table 4 and Table 5 in our paper state, "Uses all 96/52 images unless otherwise noted (K)." Therefore, the Uni MS-PS results report there correspond to the setting using **all available images** (96/52) from DiLiGenT and Luces, rather than its best performance achieved with a subset of 30 input images.
>
> Moreover, to address your concern, we present a new comparison between our method and Uni MS-PS using 30 input images in the tables below. The metric is Mean Angular Error (MAE) (lower is better). The '$\pm$' symbol separates the mean (preceding) from the standard deviation (following), based on 10 test runs.
>
> **Comparison on DiLiGenT (K=30):**
>
> | Method | Ball | Bear | Buddha | Cat | Cow | Goblet | Harvest | Pot1 | Pot2 | Reading | **Avg. MAE** |
> | :--- | :---: | :---: | :---: | :---: | :---: | :---: | :---: | :---: | :---: | :---: | :---: |
> | Uni MS-PS | 1.82±0.05 | 3.13±0.11 | 6.10±0.12 | 3.44±0.05 | **3.99±0.09** | 6.52±0.15 | 8.89±0.23 | 4.11±0.07 | 4.65±0.07 | 6.99±0.16 | 4.96±0.13 |
> | **Ours** | **1.75±0.02** | **2.63±0.07** | **6.09±0.15** | **3.53±0.05** | 4.01±0.08 | **5.28±0.10** | **8.74±0.14** | **4.10±0.05** | **4.30±0.07** | **6.75±0.15** | **4.72±0.09** |
>
> **Comparison on LUCES (K=30):**
>
> | Method | Ball | Bell | Bowl | Buddha | Bunny | Cup | Die | Hippo | House | Jar | Owl | Queen | Squirrel | Tool | **Avg. MAE** |
> | :--- | :---: | :---: | :---: | :---: | :---: | :---: | :---: | :---: | :---: | :---: | :---: | :---: | :---: | :---: | :---: |
> | Uni MS-PS | 10.29±0.08 | 10.51±0.11 | **6.77±0.19** | 12.63±0.29 | 9.60±0.34 | 13.35±0.42 | 6.27±0.15 | 8.44±0.36 | 25.46±0.44 | 6.18±0.16 | 11.38±0.38 | 15.97±0.41 | 11.27±0.24 | 11.74±0.50 | 11.10±0.36 |
> | **Ours** | **10.09±0.07** | **8.82±0.09** | 6.89±0.33 | **12.53±0.25** | **6.23±0.25** | **8.34±0.11** | **6.19±0.28** | **5.99±0.17** | **23.00±0.34** | **6.16±0.20** | **9.81±0.34** | **9.90±0.26** | **10.20±0.12** | **8.29±0.39** | **9.44±0.25** |
>
> Regarding these tables, we must clarify a few points:
> * The results of Uni MS-PS may show slight variations from its original paper because we re-run these 10 tests to compute the mean and standard deviation.
> * The deviation exists because, in a partial-input setting (e.g., 30/96), the random selection of *which* 30 images are used impacts the result. This is why our main paper (Tab. 4 & 5) used the full set of images to minimize randomness.
> * In our paper, for partial inputs (e.g., K=16, K=32), we reported the mean of 10 runs and omitted the standard deviation for visual clarity, as the variance is relatively small.
> * While Uni MS-PS achieves slightly better results on a few specific objects (e.g., 'Cow', 'Bowl'), our overall average performance remains **significantly better**. This does not affect the main conclusions of our paper.

---

> ### Author Response · Authors · 2025-11-19
>
> **Q1: Do PS-Verse Level 1 data to Level 4 data have ground truth normals?**
>
> **Yes, they do.** Our PS-Verse dataset is **synthetic data** generated using Blender, so all data (Levels 1-5) have corresponding ground truth normals. You can refer to **Fig. A4** in the updated PDF. We suspect the misunderstanding arose because we used the term "normal maps" when describing the Level 5 data. We need to clarify that "normal maps" is a 3D computer graphics technique used in Level 5 to give models the visual appearance of high-frequency geometric details, which results in more complex ground truth normals. You can refer to **Sec. A 1.3.2** for this explanation. To prevent this misunderstanding, we have standardized the terminology to "**Normal Mapping**" throughout the updated PDF. We apologize for the confusion.
>
> **Q2: For the feature similarity metric (CISM), what are the two features to compare with?**
>
> The CSIM / SSIM scores we report are the average of all pairwise computations between the features. For example, if there are six features, we compute all **$C_6^2$** pairwise comparisons and then average the results. We apologize that this was not described clearly. We have now added a description for this in the updated PDF (**L369-373**).
>
> **Q3: In table 4, for POT2, SDM-UniPS seems to achieve better results?**
>
> Thank you for pointing this out. This is a typo, and we apologize for that. We have corrected it in the updated PDF.
>
> ---
>
> Our work has already received high recognition from **the other three reviewers**, and we truly hope that after reading our rebuttal, you can also recognize the value of our work. Thank you again!

---

### Official Review · Reviewer_7ywv · 2025-10-28

**Soundness:** 3
**Presentation:** 3
**Contribution:** 3
**Rating:** 8
**Confidence:** 5

**Summary:**

The paper proposes a new encoder to address two issues in Photometric Stereo (PS): (1) the coupling between normal and lighting information, and (2) the loss of high-frequency details. To tackle the first problem, the authors introduce Light Register Tokens and an Interleaved Attention Block to decouple normal and lighting features. Furthermore, they employ a wavelet-based dual-branch structure combined with specific loss to preserve high-frequency details.

**Strengths:**

The proposed method demonstrates solid performance on both real and synthetic datasets. The ablation studies are comprehensive and well-conducted.

**Weaknesses:**

- Line 64 seems quite contradictory, should it be normal features instead?
- Could the authors provide a more physically grounded explanation for explaining the effect of feature similarity? The current discussion mainly relies on experiments illustration.
- Could the Light Token also be used for light source estimation? If so, how accurate would it be?
- It would be much better to discuss and acknowledge PS work under general setup.

**Questions:**

See above

---

> ### Author Response · Authors · 2025-11-19
>
> We sincerely thank the reviewer for the time and effort invested in reviewing our paper! Your questions are both valuable and insightful, and they have been a great help in improving the manuscript's quality. They are also very inspiring for our future research. The corresponding changes addressing your questions have been highlighted in **Blue** in the newly uploaded PDF.
>
> ---
>
> **W1: Line 64 seems quite contradictory, should it be normal features instead?**
>
> Your understanding is absolutely correct! This is a typo, and we apologize for the confusion it caused. We have corrected it in the updated PDF.
>
> **W2: Could the authors provide a more physically grounded explanation for explaining the effect of feature similarity?**
>
>
> Thank you for your insightful question! Fundamentally, Universal PS aims to solve an inverse problem: recovering the intrinsic surface normal field $N$ from a set of observations $ {I\_f}\_{f=1}^F$ formed under varying, unknown illumination conditions $ {\mathbf{L}\_f}\_{f=1}^F$. The imaging process can be formally described as $I\_f = \mathcal{F}(N, \rho, \mathbf{L}\_f)$, where $\mathcal{F}$ represents the rendering equation and $\rho$ the surface reflectance.
>
> An ideal encoder $E$ seeks to extract a unified feature representation $F\_{enc} = E({I_f}\_{f=1}^F)$ that is strictly **illumination-invariant**, effectively marginalizing out the extrinsic variable ${\mathbf{L}\_f}\_{f=1}^F$ Feature similarity (CSIM/SSIM) serves as a direct quantitative metric of this invariance; a low CSIM/SSIM score implies that the feature $F_{enc}$ retains significant dependency on $ {\mathbf{L}\_f}\_{f=1}^F$, indicating a failure to decouple extrinsic illumination from intrinsic geometry.
>
> The decoder $D$ is tasked with learning the mapping $\hat{N} = D(F\_{enc})$. When features are not decoupled, the decoder confronts a highly ill-posed problem: it receives highly variable inputs $F_{enc}$ for the exact same physical geometry $N$, which inevitably introduces ambiguity and variance into the estimator, manifesting as higher reconstruction error (MAE).
>
> Therefore, a primary driver for our LINO UniPS is the introduction of an improved encoder $E$. By explicitly employing Light Register Tokens supervised by Light Alignment to physically isolate variant illumination components ${\mathbf{L}\_f}\_{f=1}^F$, our encoder ensures that the feature representation passed to the decoder remains pure and consistent. This effectively mitigates the ill-posed nature of the decoding task, naturally yielding higher accuracy.
>
> We have incorporated this part into our paper. For a more detailed discussion, please refer to **Sec. A 1.4.1** of the newly uploaded PDF.
>
> **W3: Could the Light Token also be used for light source estimation? If so, how accurate would it be?**
>
> Your question is very insightful and inspiring! This is not currently feasible with our method. Our Light Register Tokens are supervised by Light Alignment Loss, which performs an **alignment** in a high-dimensional space (similar to methods like REPA (Representation Alignment for Generation, CVPR 2025) and VAVAE (Reconstruction vs. Generation, CVPR 2025), rather than a direct **regression** loss.
>
> However, we see great potential in this direction and will definitely explore simultaneous Photometric Stereo and light estimation in our future work.
>
> **W4: It would be much better to discuss and acknowledge PS work under general setup.**
>
> This is a nice comment! While the Universal PS paradigm offers robust generalization capabilities, it is essential to acknowledge specific inherent drawbacks when contrasted with traditional approaches under a general setup.
>
> * Traditional Calibrated PS methods use explicit, known light source parameters. In contrast, Universal PS lacks any such explicit light source input during inference (although we employ Light Register Tokens to mitigate the absence of this information). This difference means that Universal PS methods inherently struggle to unambiguously distinguish whether light originates from "above" or "below" the surface when handling near-planar objects, whereas Calibrated PS handles these cases relatively better.
> * The Universal PS paradigm relies on massive training data. Conversely, traditional Calibrated and Uncalibrated PS approaches do not rely as heavily on large-scale training datasets.
>
> We have now added this discussion to the appendix of the updated PDF. For a more detailed analysis, please refer to **Sec. A 1.4.2**.

---

### Official Review · Reviewer_N2Ck · 2025-10-30

**Soundness:** 4
**Presentation:** 3
**Contribution:** 4
**Rating:** 8
**Confidence:** 4

**Summary:**

In this paper, the authors introduce a ViT-based framework, named LINO UniPS, for universal photometric stereo. They target at tackling two major challeneges in universal photometric stereo, namely 1) ineffective decoupling of illumination and normal cues and 2) loss of geometry details. Specifically, they introduce Light Register Tokens and an Interleaved Attention Block to explicitly decouple illumination from normal features, yielding a unified feature representation. They also adopt a Wavelet-based Dual-branch Architecture and a Normal-gradient Perception Loss that substantially improve the reconstruction of fine-grained geometric details. Further, they build a synthetic dataset, named PS-Verse, graded with surface complexity and lighting diversity to support similar research.

**Strengths:**

+ The introduction of three types of Light Register Tokens for aggregration of illumination information of three different illumination types sounds logical and novel. It helps improve the decoupling of illumination from normal features. Its effectiveness has been demonstrarted in ablation study.
+ The proposed light alignment supervision sounds logical and novel. It helps Light Register Tokens learn to capture the respective illumination information. Its effectiveness has been demonstrated in ablation study.
+ The Interleaved Attention Block introduces a global cross-image attention mechanism. Although global attention is not a novel idea, but the four interleaved attention layers allow aggregrating information across multiple hierarchical levels and help better decoupling illumination and normal features. The effectiveness of the Interleaved Attention Block has been demonstrated in ablation study.
+ The Wavelet-based Dual-Branch Architecture sounds logical and novel. It helps preserve details throughout the network. Its effectiveness has been demonstrated in ablation study.
+ The Normal Gradient Perception Loss sounds logical and novel. It helps to enhance high-frequency areas. Its effectiveness has been demonstrated in ablation study.
+ The large-scale synthetic dataset PS-Verse can benefit further research in universal photometric stereo.
+ SOTA results have been reported on DiLiGenT and Luces datasets.

**Weaknesses:**

- The figures and captions need further improvement. For instance, the pipeline in fig. 2 is rather complicated and difficult to understand. It does not match well with the detailed description of the modules. What do the different colors represent? In fig. 3, it is not clear how to interpret the attention maps for the different Light Register Tokens. More detailed discussions are needed to better understand how the figures demonstrate the effectiveness of the different Light Register Tokens.
- The concept of inter-image and intra-image context in this paper is rather confusing. In lines 228-229, it mentioned that frame attention captures inter-image context. Should the frame attention capture intra-image (within-image) context while the global attention capture inter-image (between-images) context instead? Discussions in the Appendix also show the same confusion.
- The ablation study has demonstrated the effectiveness of each proposed component quantitatively. It would, however, be also important to show the corresponding qualitative results to enable readers to visually perceive the effecti of each component.
- Even with the details provided in the Appendix, it is not sufficient to reproduce the results. For instance, in lines 707-714, the Light Register Tokens have a dimension of C, but they are concatenated with the wavelet/downsample tokens with a dimension of D. In lines 751-755, should the "1 - " goes inside the summation sign? Otherwsie, what is the purpose of including "1" in the loss? In lines 808-814, the steps in transformating the aggregated features into the four-level feature pyramid is not very clear.

**Questions:**

- Theoretically, the wavelet transform already includes a downsampled version of the image. Why is it necessary to include a naive downsampling branch in parallel?
- Three Light Register Tokens are introduced specifically for three illumination types, namely point, direction, and environment lights. Have the authors consider using a single generic light register token instead? Does this design generalize well to other light representations?

**Details Of Ethics Concerns:**

None.

---

> ### Author Response · Authors · 2025-11-19
>
> We sincerely thank the reviewer for the time and effort dedicated to this review. We are extremely grateful for your high recognition of our work! Your insightful and constructive comments have been critical in helping us improve the quality of our manuscript. We have highlighted the corresponding changes addressing your questions in **Red** in the updated PDF.
>
> ---
> **W1: The figures and captions need further improvement.**
> 1.  **Complexity of Fig. 2:** Thank you for your comment! We acknowledge that Fig. 2 was indeed a bit complicated. Our original intention with the colors was to distinguish between different modules and tokens. To address this, we have redrawn Fig. 2 for better clarity. Please see the **Fig. 2** in the updated PDF.
> 2.  **Interpretation of Fig. 3:** Thank you for the suggestion! We have added a more detailed discussion to clarify how these maps demonstrate the effectiveness of the different tokens. Please refer to the **caption for Fig. 3** and the corresponding discussion (**L202-211**) in the updated PDF.
>
> **W2: The concept of inter-image and intra-image context in this paper is rather confusing.**
>
> Thank you for your sharp-eyed catch! Your understanding is exactly right. Frame Attention captures intra-image (within-image) context, while Global Attention captures inter-image (between-images) context. We have corrected this typo in both the main text and the appendix of the updated PDF.
>
> **W3: It would be important to show the corresponding qualitative results.**
>
> Thank you for pointing this out; it is critical for demonstrating the efficacy of our method. We have now added the qualitative results for the ablation study. Please refer to **Fig. 6** in the updated PDF.
>
> **W4: Issues in the Appendix Details.**
>
> 1.  **Dimension of Light Register Tokens:**
> This is a typo. The dimension of the Light Register Tokens should be $D$. We have corrected this in the updated PDF and apologize for the confusion.
> 2.  **The "1 -" in the loss function:**
> We argue that this '1' should be placed outside the summation symbol and must be retained. This is because our Light Alignment Loss is defined as a **cosine similarity loss**. Its standard mathematical formulation is: $ \mathcal{L} = 1 - \sum_{i=1}^{n} A_i B_i $, which includes the '1' outside the summation symbol, consistent with our implementation.
> 3.  **Transformation steps for the feature pyramid:**
> Thank you for pointing this out! We have further polished the text description. Additionally, we have added a new figure to the appendix showing the pipeline and the corresponding tensor shape transformations during the forward pass to help clarify the details. Please refer to **L873-890** and **Fig. A1** in the updated PDF.
>
> **Q1: Theoretically, the wavelet transform already includes a downsampled version of the image. Why is it necessary to include a naive downsampling branch in parallel?**
>
> This is an excellent question! In fact, we conducted experiments on this and concluded that the **naive downsampling branch is necessary**, as the dual-branch architecture showed a clear performance improvement over a Wavelet-only approach. We think the reason is that the two branches are **complementary**: The Wavelet Branch focuses on explicitly preserving high-frequency, fine-grained geometric details. The Naive Downsample Branch, while lossy in detail, provides a smoother, more anti-aliased low-frequency representation, offering robust support for global image-domain semantic information.
>
> The fusion of these complementary advantages allows the model to achieve superior accuracy. This is why we retained the parallel naive downsample branch despite the wavelet transform's low-frequency component. We have included this experiment in **Sec. A.1.6.3** of the updated PDF.
>
> **Q2: Have the authors considered using a single generic light register token instead?**
> This is a valuable question! We run an additional ablation study to test this, and our conclusion is that the **generic token method performs worse than our specialized token method**. Our analysis suggests that a generic token is forced to entangle physically disparate lighting cues (e.g., high-frequency point lights vs. low-frequency environment lights), which degrades the subsequent normal prediction. We have added this experiment to **Sec. A.1.6.2** of the updated PDF. Please refer to the new file for detailed analysis and results.
>
> **Q3: Does this design generalize well to other light representations?**
>
> Yes, this design generalizes well to the vast majority of lighting scenarios. This is because Point, Direction, and Environment lights act as **'primitives'** that, in combination, can represent most complex lighting conditions.
>
> This generalization capability is demonstrated by the fact that our model, trained purely on synthetic data, performs exceptionally well on real-world, in-the-wild data. Please refer to the qualitative results in **Fig. 1, Fig. 7, Fig. A10, and Fig. A12** in the updated PDF.

---

### Comment · Area_Chair_gp7Z · 2025-11-26

Dear reviewers,

Please check the author's reply. Feel free to raise any questions or start a discussion, regardless of whether you will change the score.

Your AC.

---

### Meta-Review · Area_Chair_K9eP · 2026-01-10

**Summary:**

This paper proposes a ViT-based universal photometric-stereo system that explicitly decouples lighting from surface normals while preserving high-frequency geometry. The core ideas are 1) Light Register Tokens with light-alignment supervision to isolate point, directional, and environment illumination, 2) an Interleaved Attention Block that performs cross-image aggregation so the encoder can factor out lighting, and 3) a wavelet-based dual branch plus normal-gradient loss to retain fine details. The paper also introduces PS-Verse, a large synthetic dataset graded by geometric and lighting complexity, and reports new state-of-the-art results on DiLiGenT and Luces with improved generalization to in-the-wild data

Here are the main concerns and how the rebuttal addressed them.

1) Clarity and architectural confusion.
Several reviewers found Fig. 2/3 and the inter- vs. intra-image attention roles confusing. The authors redrew Fig. 2, expanded Fig. 3 captions, corrected the frame/global attention typo, and added a tensor-shape pipeline in the appendix, materially improving interpretability.

2) Whether the design choices are necessary (tokens, branches).
Reviewers questioned the need for three specialized light tokens and for a naive downsample branch alongside wavelets. The rebuttal added targeted ablations showing a generic token performs worse and that the dual-branch outperforms a wavelet-only model because low-frequency robustness complements high-frequency detail.

3) Physical grounding of feature decoupling.
One reviewer asked for theory beyond empirical CSIM/SSIM. The authors provided a physically grounded inverse-rendering argument: decoupled features reduce decoder ill-posedness by removing illumination variance, explaining the observed MAE gains.

4) Fairness and correctness of comparisons.
A low-score review alleged missing best results from Uni MS-PS and unclear ground truth. The rebuttal clarified that the main tables use all available images, added fair K=30 re-runs where LINO UniPS wins on average, fixed typos, and explained that PS-Verse is fully synthetic with ground-truth normals at all levels.

5) Generality, efficiency, and missing ablations.
New qualitative ablations, token-type ablations, and in-the-wild multi-object results were added. The authors also clarified that frame tokens are not duplicated, so the three light registers add negligible compute.

Overall, the rebuttal directly resolved the technical and factual objections, strengthened the empirical case, and improved presentation.

Recommendation: Accept.

**Reviewer Concerns:**

Here are the main concerns and how the rebuttal addressed them.

1) Clarity and architectural confusion.
Several reviewers found Fig. 2/3 and the inter- vs. intra-image attention roles confusing. The authors redrew Fig. 2, expanded Fig. 3 captions, corrected the frame/global attention typo, and added a tensor-shape pipeline in the appendix, materially improving interpretability.

2) Whether the design choices are necessary (tokens, branches).
Reviewers questioned the need for three specialized light tokens and for a naive downsample branch alongside wavelets. The rebuttal added targeted ablations showing a generic token performs worse and that the dual-branch outperforms a wavelet-only model because low-frequency robustness complements high-frequency detail.

3) Physical grounding of feature decoupling.
One reviewer asked for theory beyond empirical CSIM/SSIM. The authors provided a physically grounded inverse-rendering argument: decoupled features reduce decoder ill-posedness by removing illumination variance, explaining the observed MAE gains.

4) Fairness and correctness of comparisons.
A low-score review alleged missing best results from Uni MS-PS and unclear ground truth. The rebuttal clarified that the main tables use all available images, added fair K=30 re-runs where LINO UniPS wins on average, fixed typos, and explained that PS-Verse is fully synthetic with ground-truth normals at all levels.

5) Generality, efficiency, and missing ablations.
New qualitative ablations, token-type ablations, and in-the-wild multi-object results were added. The authors also clarified that frame tokens are not duplicated, so the three light registers add negligible compute.

**Reviewer Scores:**

N2Ck (8): likely +0 — already strong accept; clarity and ablations further reinforce.

7ywv (8): likely +0 — main requests (physical grounding, PS-setup discussion, typos) were satisfied.

nvYr (8): likely +0 — added qualitative ablations and token analyses address remaining questions.

zyYx (2): likely +2 to +3 — key objections (fair Uni MS-PS comparisons, ground-truth normals, multi-object capability, metric definitions) were corrected with new experiments and clarifications, moving this from outlier reject toward borderline/weak accept.

---

### Decision · Program_Chairs · 2026-01-26

Accept (Poster)